# Transcriptomic profiling and machine learning uncover gene signatures of psoriasis endotypes and disease severity

Ashley Rider[1,16], Henry J. Grantham [1,16], Graham R. Smith [2,16], David S. Watson [3,4,16], John Casement[2], Simon J. Cockell [2,5], Jack Gisby[3], Amy C. Foulkes[6], Rafael Henkin [3], Wasim A. Iqbal[2], Tom Ewen [1], Shoba Amarnath [5], Sandra Ng [3], Paolo Zuliani [7,8], Nick Dand [9], Deborah Stocken[10], Christopher Traini[11], Elizabeth Thomas[11], Shanker Kalyana-Sundaram[11], Deepak K. Rajpal[11,12], Kathleen M. Smith [13], Jonathan N. Barker[14], Christopher E. M. Griffiths[6], Paola Di Meglio [14], Catherine H. Smith [14], Richard B. Warren[6], Michael R. Barnes [3,17] ✉ & Nick J. Reynolds [1,15,17] ✉ On behalf of the PSORT consortium*

## Abstract

**Background** Despite increased understanding of psoriasis pathogenesis, molecular classification of clinical phenotypes and disease severity is poorly defined. Knowledge gaps include whether molecular endotypes of psoriasis underlie distinct clinical phenotypes and the positive and negative molecular regulators of disease severity across tissue compartments. **Methods** We performed comprehensive RNA sequencing of skin and blood (n = 718) from prospectively-recruited, deeply-phenotyped discovery and replication cohorts of 146 subjects with moderate-to-severe chronic plaque psoriasis initiating TNF-inhibitor (adalimumab) or IL-12/23-inhibitor (ustekinumab) therapy. **Results** Here we show, using two complementary dimensionality reduction methods, that co-expressed gene modules and factors within skin and blood are significantly associated with psoriasis phenotypes and disease severity. We identify a 14-gene signature negatively associated with BMI in nonlesional skin and with disease severity in lesional skin. Genotype integration reveals that HLA-DQA1*01 and HLA-DRB1*15 genotypes are positively associated with baseline psoriasis severity. Using explainable machine learning models, we define two disease severity-associated gene modules in lesional skin - one positive, one negatively-associated - and a 9-gene signature in lesional skin predictive of disease severity. Disease severity signatures in blood are only seen following adalimumab exposure, suggesting greater systemic impact of adalimumab compared to ustekinumab, in line with its side effect profile. In contrast, a gene signature in blood linked to HLA-C*06:02 status is independent of disease severity or drug. **Conclusions** These findings delineate gene-environmental and genetic effects on the psoriasis transcriptome linked to disease severity.

## Plain language summary

Psoriasis is a common and debilitating skin disease, linked to other inflammatory conditions. A lot is known about what causes psoriasis and the factors that influence it, but doctors still cannot offer personalised treatments. This is because it has been difficult to understand what makes psoriasis more or less severe, why people respond differently to treatment, or why some people develop related diseases. To help address this, we collected skin and blood samples and personal information from people with severe psoriasis across the United Kingdom. Using computer-based methods, we found shared biological processes that link the disease with obesity and help predict its severity.

Psoriasis is a common, multifaceted immune-mediated inflammatory disease (IMID) characterised by symmetrical erythematous, hyperplastic and scaly plaques affecting the skin, with associated systemic inflammatory disorders including psoriatic arthritis, cardiovascular disease and metabolic syndrome, which contribute to premature mortality[1]. The aetiology and pathophysiology of psoriasis is complex and multifactorial[1]. Over the last two decades significant progress has been made in understanding the pathophysiology of psoriasis and the contributions of various factors,

A full list of affiliations appears at the end of the paper. *A list of authors and their affiliations appears at the end of the paper. ✉e-mail: m.r.barnes@qmul.ac.uk; nick.reynolds@newcastle.ac.uk

including: genetic predisposition[2]; environmental factors including infection, trauma and diet; and acquired immune (e.g. T helper 1 (Th1), T17 cells, IL-17, IL-23 cytokines) and innate autoinflammatory factors (e.g. TNF, IL-36) which represent targets for highly effective biologic therapies. However, less progress has been made in translating these advances into individualised patient care. In part, this relates to significant knowledge gaps along the translational pathway that include: (a) whether molecular endotypes within clinically homogeneous stable plaque psoriasis underlie distinct clinical phenotypes (e.g. sub-groups of subjects with specific comorbidities), (b) the positive drivers and negative molecular regulators of disease severity across tissue compartments, and (c) the relationship between molecular endotypes and clinical response to therapy, including the side effect profiles of targeted therapies.

To address these questions, Psoriasis Stratification to Optimise Relevant Therapy (PSORT), an academic-industrial UK stratified medicine consortium[3], prospectively recruited formally-powered, deeply-phenotyped discovery and replication psoriasis patient cohorts during the early phase of treatment with two distinct biologics, adalimumab (TNF inhibitor) and ustekinumab (IL-12/23 inhibitor) (Fig. 1a, b). Utilising this large multiomic dataset, we aimed to identify gene networks linked to specific disease endotypes, defined by clinical and phenotypic features measured at baseline (e.g. BMI), and to disease severity endotypes, defined by Psoriasis Area and Severity Index (PASI)-associated gene expression profiles (Fig. 1c, d).

Gene expression is influenced by multiple factors across different cell types. Network analysis of bulk RNA-seq data from large clinical cohorts can capture coordinated patterns of expression by identifying underlying latent factors and co-expression modules. Moreover, these gene networks can more accurately represent key biological processes and generate signatures for disease classification and therapeutic response prediction, compared with analyses of individual genes[4]. We therefore hypothesised that such network signatures would map to specific clinical phenotypes of psoriasis, including disease severity over time in response to biologic therapy.

Our integrative multiomic analysis across lesional and nonlesional skin and whole blood identifies gene signatures in distinct tissue compartments and cell types that classify psoriasis endotypes and associate with disease severity. These signatures highlight both pathogenic pathways active in psoriasis and systemic immune processes detectable in blood, distinguish disease endotypes linked to genetic factors, and identify reproducible biomarkers of disease severity. Together, these findings provide a framework for understanding the molecular heterogeneity in psoriasis.

## Methods
### Prospective observational study
This study included 146 subjects with moderate to severe chronic plaque-type psoriasis (PASI > 10) recruited prospectively into the Psoriasis Stratification to Optimise Relevant Therapy (PSORT) study at 6 centres in the UK between May 2015 and May 2018 and due to start biologic therapy (ustekinumab or adalimumab) as part of routine clinical practice[3]. Exclusion criteria included use of systemic or biologic treatments in the two weeks prior to study entry (or four x t½ of last treatment, whichever was longer), use of PUVA therapy for 3 months or UVB for 1 month prior to study entry or use of topical treatments at the site of biopsies (except for emollients) for 2 weeks prior to study entry, as well as serious/uncontrolled systemic disease. We studied 89 psoriasis subjects within the discovery cohort and replicated findings in a further cohort of 57 subjects. For the replication cohort, samples for RNA sequencing were selected from patients whose treatment response (PASI 50/75/90 and non-responders) broadly matched those in the earlier discovery cohort. Selection was based solely on clinical response and did not consider molecular or demographic data. Subjects commencing adalimumab by subcutaneous injection received 80 mg at baseline, then 40 mg at week 1, then 40 mg every 2 weeks as per label and those starting ustekinumab received 45 mg or 90 mg according to body weight, as per label. Participants self-administered doses that fell between study visits and the time and date of these were recorded in the Case Report Form. The Psoriasis Association provided Patient and Public Involvement

and Engagement, which influenced the study design. The study was conducted in accordance with the declaration of Helsinki, was approved by the London Bridge research ethics committee (REC reference: 14/LO/1685PSORT) and subjects provided written informed consent.

Patients completed detailed demographic questioning, including reporting information on comorbidities and concomitant and previous medication. Disease severity and response to therapy were assessed using the PASI, Physician Global Assessment (PGA) and DLQI. Clinical samples, including blood and lesional skin punch biopsies (edge of psoriasis plaque with site preference for lower back or buttock), were collected under local anaesthetic at baseline, one week (prior to the second injection of adalimumab) and 12 weeks of treatment. Biopsies were derived from the same body sites at each time point and, if possible, lesional biopsies were taken from the same plaque as baseline. Non-lesional skin with a minimum distance of 2 cm from the edge of nearest plaque was also collected at baseline and week 12 (and a minimum distance of 2 cm between initial and subsequent biopsies) and a further blood sample was taken at 4 weeks. Patients had been identified in line with recommendations for initiation of biologic therapies in the UK. Screening investigations had been completed prior to recruitment. Adverse events were recorded but did not form part of primary analysis.

### Power calculation
Based on our affiliated pilot investigation[5], using the method of Guo et al.[6], we calculated the requisite sample size to achieve 90% power to detect differential expression associated with response. Imposing a 5% FDR threshold and a target log fold change of 1.5, we determined that a study would require 40 subjects to achieve 90% power to identify transcriptomic markers of biologic response for patients with chronic plaque psoriasis. Power curves projected across an expected range of fold changes at 1% and 5% differential expression in Supplementary Fig. 1.

### RNA extraction and quality control
**Skin samples.** RNA was preserved in the skin punch biopsies using RNAlater Stabilization Solution (Invitrogen AM7022). Biopsies were stored at 4 °C in RNAlater overnight and the solution removed prior to long term storage at −80 °C (according to the manufacturer's instructions).

Biopsies were transferred into pre-cooled 2 ml lysing tubes (Precellys, CK mix) containing lysis buffer (10 µL 2-ME/mL RLT Plus) supplied in the Qiagen AllPrep DNA/RNA Kit (Cat. No. 80204). Homogenisation was performed in the TissueLyser LT (Qiagen Cat. No. 85600) over 10 ×2-min cycles at 50 Hz. Samples were cooled on wet ice for one minute between cycles. Tissue debris was pelleted at 13,000 RPM for 3min, and the supernatant was transferred to an Allprep DNA spin column. DNA/RNA was then extracted following the Qiagen AllPrep kit's protocol. RNA concentration/integrity was checked using the Agilent Bioanalyzer 2100 with the RNA 6000 Nano assay (Agilent: 5067-1511); only samples with an RNA integrity number (RIN) of 8 or more were sequenced.

**Blood samples.** RNA was isolated from human whole blood collected in PAXgene blood RNA tubes (Qiagen #762165) utilising the QIAsymphony SP (Qiagen #9001297) with the QIAsymphony PAXgene Blood RNA kit 96 (Qiagen #762635). The manual processing of the whole blood samples was performed following the manufacturer's protocol and loaded onto the QIAsymphony SP. The RNA isolation protocol implemented was a custom protocol based on the standard automation protocol PAXgene_RNA_V5.xml. The custom protocol is PAXRNA_CR22332_2915.xml and contains the following modification to the standard protocol "elution buffer taken out of accessory trough. Accessory trough will be displayed as ETOH on the touch screen." The elution buffer used was Invitrogen UltraPure DNase/RNase-free distilled water #10977015. Whole blood RNA samples were eluted into 80 µL of Invitrogen UltraPure DNase/RNase-free distilled water #10977015 and plated into 8×12 elution plates. RNA quality control for quantity was performed with Qubit RNA Broad Range (BR) (ThermoFisher Scientific #Q10211) on a Molecular Devices Gemini plate reader following the

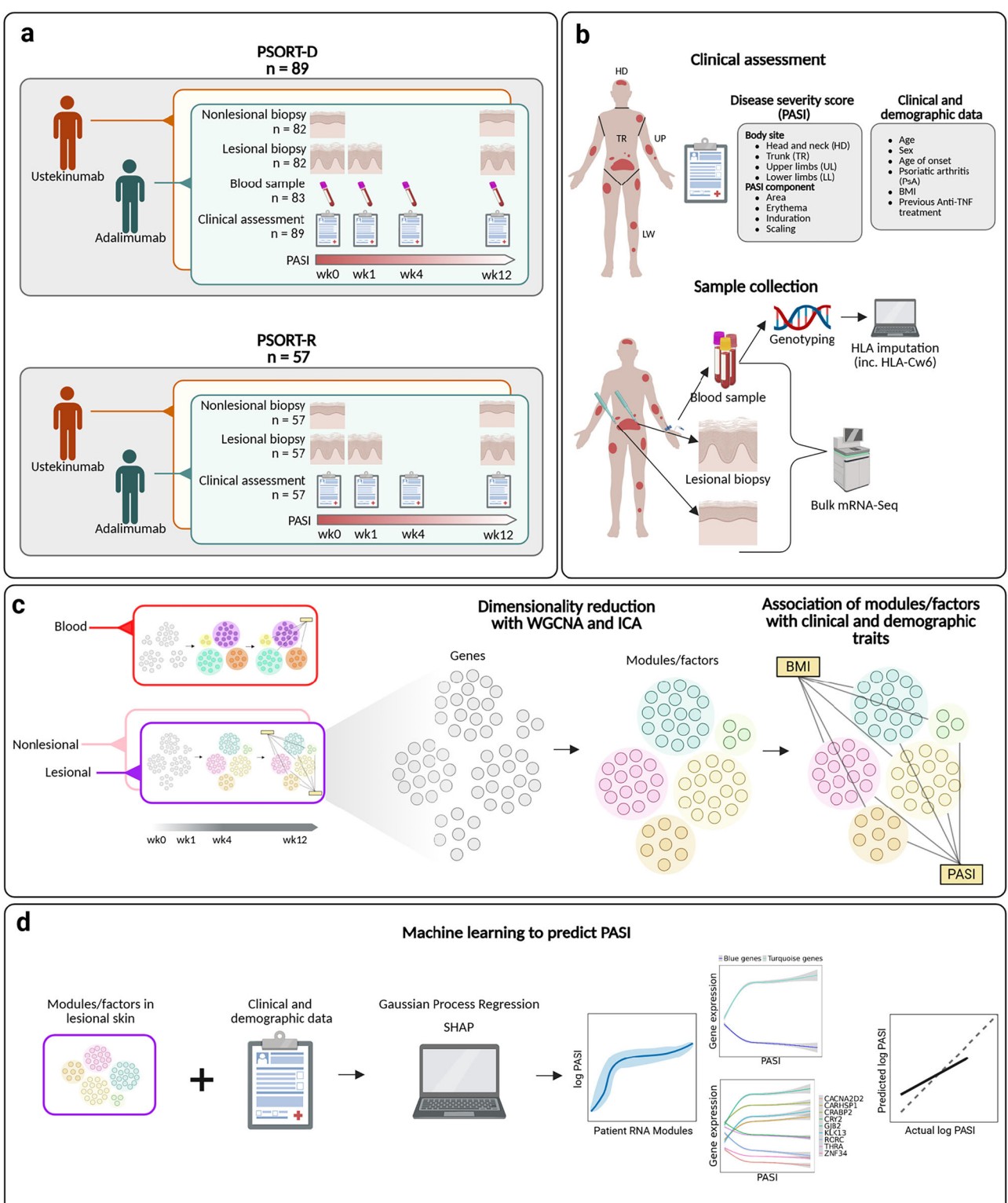

manufacturer's protocol for reagent and sample preparation. RNA quality control for integrity was performed with an Agilent TapeStation 4200 (Agilent #G2991BA) using the RNA Screentape assay (Agilent #5067-5576,77,78) following the manufacturer's protocol.

### RNA sequencing

The sequencing libraries for the PSORT-D skin samples were prepared using the Illumina Truseq stranded mRNA kit and sequenced on an Illumina HiSeq 3000 with 2x101bp read length. The sequencing libraries for the PSORT-D blood samples were prepared from total RNA using the Kapa mRNA HyperPrep kit and depleted of rRNA and globin mRNA using the QIAseq FastSelect RNA Removal Kit by Qiagen; sequencing was done on an Illumina HiSeq 4000 using 2x75bp read length. The sequencing libraries for the PSORT-R skin samples were prepared using the Illumina Truseq stranded mRNA kit and sequenced on an Illumina NovaSeq 6000 with 2x250bp read length. As the approved participant consent forms did not

**Fig. 1 | Summary of study design, including patient recruitment, sample collection and analysis methodology. a, b** Participants with moderate-to-severe psoriasis eligible for biologic therapy were enroled into discovery (PSORT-D, *n* = 89) and replication (PSORT-R, *n* = 57) cohorts and treated with either adalimumab or ustekinumab. At baseline and follow-up visits (weeks 0, 1, 4 and 12), lesional kin, nonlesional-skin and peripheral blood samples were collected, and PASI scores obtained. **c** To discover disease endotypes, we applied two complementary dimension-reduction methods separately to skin and blood RNA-seq data. Weighted Gene Co-expression Network Analysis (WGCNA) groups transcripts with similar expression into *modules* and represents each module by its first principal component (the module eigengene). Independent Component Analysis (ICA) decomposes the dataset into statistically independent latent factors that capture underlying biological signals. Eigengenes and latent factors were correlated with clinical traits (e.g. BMI, HLA-C*06:02 genotype) and with PASI across time-points to reveal both baseline (disease) and disease-severity signatures. **d** Finally, we trained a Gaussian Process Regression (GPR) model—a flexible non-linear machine learning approach—using eigengenes and latent factors to predict log-transformed PASI. Model interpretability was provided by SHAP (SHapley Additive exPlanations), which quantifies the contribution of each module or factor for example to the prediction, allowing derivation of a concise gene signature of disease severity. PASI Psoriasis Area and Severity Index, BMI body-mass index, LS lesional skin, NLS nonlesional skin, WGCNA weighted gene co-expression network analysis, ICA independent component analysis, GPR Gaussian Process Regression, SHAP SHapley Additive exPlanations. Created in BioRender. Rider, A. (2025) https://BioRender.com/7qdytt1.

include permissions for the sharing of personally identifiable raw sequencing data, we provide raw and adjusted gene count data from our RNA-seq analysis. The count data are available at Array Express under accession number E-MTAB-14509. This approach allows us to share valuable processed data for replication, validation and further analysis while respecting participant privacy and adhering to governance and ethical guidelines.

### Genotype data and HLA imputation
DNA was isolated from blood using standard methods. Genotyping was performed with Illumina HumanOmniExpressExome-8 v1.2 and v1.3 BeadChips, followed by quality control with standard tools, as previously described[7]. *HLA-C*06:02* (HLA-Cw6) genotype was imputed using SNP2HLA (version 1.0.3) based on the Type 1 Diabetes Genetics Consortium reference panel[8].

### Genomic and transcriptomic data analysis workflow
Analysis was conducted in the R statistical computing environment (R Core Team, 2021). Sample data analysis scripts can be found on our GitHub repository (https://github.com/C4TB/PSORT), along with extended supplemental markdown documents.

A graphical summary of the samples used for analysis is available in supplementary fig. 21. Several blood samples were RNA sequenced but identified as technical failures based on exploratory analysis and so were excluded. Following RNA sequencing of all samples, reads were pseudo-aligned using Kallisto[9]. Transcript counts aggregated gene-wise and were TMM normalised prior to modelling[10] and transformed to the $log_2$-CPM scale. An expression filter was applied to ensure that a gene has at least one count per million (CPM) in at least 5% of all libraries, leaving 16,172 genes.

### Exploratory data analysis
Principal component analysis identified tissue, time, and disease activity as the key drivers of transcriptome variation (Supplementary Fig. 22c), with relatively limited influence of demographic and other factors previously associated with therapeutic response.

### Differential expression analysis
Differential expression was tested using heteroskedastic linear models and empirical Bayes shrinkage as implemented by the voom function in the limma software package[11]. We compute *q*-values for each differential expression test using Storey's method[12], with a false discovery rate (FDR) threshold of 5%.

We built separate models to test a number of related hypotheses. We have two primary goals for this portion of the experiment[1]: to identify genes that associate with disease phenotype (e.g BMI) and[2] to identify genes that correlate with PASI irrespective of time points. We refer to these as the disease, and disease severity endotypes, respectively.

**Disease severity model.** Disease activity is measured at each time point by PASI. Building on the design of our trial study[5], we split the data by tissue and analysed samples from both treatment arms with coefficients for each drug. We accounted for repeated observations using the *duplicateCorrelation* function, which approximates a mixed model design in which patient ID is treated as a random effect. Because preliminary investigations suggested that gene expression is often a non-monotonic function of PASI, we expanded the model using a cubic spline basis of degree 3. An intercept term was included for each drug and the spline coefficients were also allowed to vary depending on the drug, except for the model underlying the volcano plot in Fig. 6, which was independent of drug. In the lesional skin and nonlesional skin models, an intercept for the Cohort (Discovery/Replication) was included. In addition to the q-value criterion, we define a signed fit range equal to the minimum to maximum range of the fitted log2CPM expression, with the sign given by the sign of the gradient at PASI = 0. In the Venn Diagrams and Volcano plots of Figs. 6, 7, we include only those genes that were assigned to a WGCNA module (see the section "Identification of gene coexpression modules" below).

### Dimensionality reduction
WGCNA and ICA decompose transcriptomes of many thousands of gene transcripts into a dataset comprised of a much smaller number of gene modules, overcoming the limitations inherent in gene-level analysis, including lower signal-to-noise ratios and a higher multiple testing burden[13], enhancing the statistical power to detect true endotype associations. These methods also reduce data complexity, offering a more holistic view of biological pathways and networks by focusing on co-expressed genes organised mutually exclusively into modules (WGCNA) or independent components (ICA), which reduces high-dimensional data into a smaller set of latent variables (or factors), with the objective of describing unobserved processes that explain patterns in gene expression, allowing for genes to belong to multiple pathways. Each module is represented by an eigengene and each factor by a metagene. Eigengenes and metagenes represent summary expression values for modules and factors, respectively. These approaches not only provide robustness against noise but also reveal biologically relevant patterns and potential novel mechanistic insights which may be missed in individual gene-level analyses.

Whereas individual genes can only be assigned to one WGCNA module, ICA allows single genes to be weighted across multiple factors in both up or down regulated states, reflecting the involvement of genes in multiple shared biological processes (Fig. 1d). These approaches are complementary, with the former being more explainable and the latter more likely to represent biological complexity.

Within skin and blood, all samples were used for WGCNA and ICA, i.e. both lesional and nonlesional samples (within skin), both drug cohorts and all time points.

### Identification of gene coexpression modules
The following steps were carried out for the PSORT-D skin and blood data separately in order to identify co-expressed gene modules in each tissue compartment. Prior to running WGCNA, the gene-level counts were filtered using a threshold that required at least one CPM in at least *n/k* libraries, where *n* equalled the number of samples and *k* equalled the number of

unique combinations of tissue type, drug, and time point. The counts were then normalised using the TMM method and transformed to log2-CPM. Selection of the appropriate soft-thresholding power β was done by plotting the values 1–20 against $R^2$, a measure of scale-free topology, and mean connectivity. The lowest value which reached the $R^2$ threshold of 0.8 was chosen; a β of 12 was chosen for skin and 5 for blood (Supplementary Fig. 23). The *blockwiseModules* function was then used to partition the genes into co-expressed modules. In brief, this function calculated the Pearson correlations between each pair of genes and raised these estimates to the selected β power in order to amplify the differences between high and low correlations. These correlations were then used to generate a topological overlap matrix (TOM) and hierarchical clustering of this matrix was used to group genes with similar expression profiles into modules. Parameters to *blockwiseModules* included a minimum module size of 30, a dendrogram cut height (for merging of similar modules) of 0.1, and the use of a signed network so that the correlations between genes were scaled to lie between 0 and 1.

## Module-trait correlations

The *moduleEigengenes* function was used to derive eigengene values for the skin and blood modules in every sample. An eigengene represents a summary expression score for a module and is analogous to the first principal component of the expression matrix for that module. Although module identification was not carried out in PSORT-R, the module assignments in PSORT-D were used to derive eigengenes for these modules in PSORT-R as well. Pearson correlation (with pairwise complete observations) was used to identify associations between modules and traits of interest. These included disease traits at baseline: age of onset, onset type (early/late), anti-TNF naïve status (Y/N), PsA (positive/negative), sex, age, BMI, and HLA-Cw6 status (positive/negative); and PASI across time in each drug cohort. Binary traits were encoded as one and zero. In skin, the module-trait correlations were carried out separately for the lesional and nonlesional samples. Significant correlations were defined by FDR ≤ 0.05; in skin, replicable correlations were defined by FDR ≤ 0.05 in the discovery cohort and nominal p-value ≤ 0.05 in the replication cohort and correlation of the same sign in both cohorts. Only traits with at least one significant correlation are displayed in the module-trait correlation heatmaps.

## Independent component analysis

Independent component analysis (ICA) was applied to identify latent variables separately in both skin (discovery cohort) and blood expression data. In each case, we included samples from both treatments and all timepoints, and we centred the data prior to factor analysis. We used the "imax" method implemented in the ica R package[14], using the maximally stable transcriptome dimension (MSTD) approach to select the optimal number of factors to compute[15] implemented in the ReducedExperiment package[16]. The number of factors recommended by MSTD was 24 for the skin expression data and 21 for blood. In order to validate the identified skin signatures, we projected the expression data from the replication cohort into the factor space defined in the discovery cohort. As a result, the feature loadings (i.e., the source signal estimates) for the skin discovery and replication cohorts are equal, permitting the investigation of the same factors in each cohort.

Factor metagenes were calculated by taking the scaled values of the estimated mixing matrix (Supplementary File Factor tables). These factor metagenes were then associated with phenotype using the same modelling approach as we employed for the module eigengenes (see Module-trait correlations, above). For factors, we additionally calculated correlations with HLA genotypes and baseline PASI. This was carried out using the combined discovery and replication cohorts following batch effect correction using limma's removeBatchEffect function.

For analyses that required a defined set of genes, we selected a set of highly aligned genes for each factor based on their loadings (Supplementary Data 5s). We selected genes with loadings that exceeded a threshold. By default, we defined this threshold at half the maximal loading for that factor

(Supplementary Data 5). For some analyses (functional enrichment analysis, BMI associations) we used a more relaxed threshold of 5; where this method resulted in the selection of less than 20 genes, we instead extracted the top 20 features.

## Deconvolution

An abundance of cell types was inferred using the CibersortX online tool[17]. To infer cell types in skin, a single-cell reference matrix was generated using single-cell RNA sequencing data from 38,274 skin cells across five inflammatory skin conditions, including psoriasis[18] downloaded from the Single Cell portal developed by the Broad Institute of MIT and Harvard (https://singlecell.broadinstitute.org/single_cell). Due to memory limitations imposed by the CibersortX tool, the size of the reference matrix was reduced by downsampling to a maximum of 200 cells per cell-type. The reference matrix was used to generate a Signature Matrix file using recommended CibersortX settings. Bulk RNA-seq data for both PSORT discovery and replication cohorts were used to generate the mixture file. Cell fractions were imputed in absolute mode using 'B-mode' batch correction.

To infer cell types in blood, the LM22 signature matrix provided by CibersortX was used. The mixture file was generated from PSORT blood RNA-seq data. Cell fractions were imputed using recommended CibersortX settings.

## Correlations of cell type fractions with latent factors and module eigengenes

Per-sample imputed absolute cell fractions for each cell type were tested for association with per-sample WGCNA module eigengene loadings and with per-sample Latent Factor loadings using the cor.test function from the R statistical computing environment. Correlation tests used Kendall's tau coefficient. Adjusted p-values were calculated with the p.adjust function using the method of Benjamini & Hochberg.

## Pathway analysis

Functional analysis of systems-level upstream regulators responsible for observed differential gene expression related to response was performed using the Upstream Regulator function in Ingenuity Pathways Analysis[19], using all genes with nominal response $p ≤ 0.05$ as input. For all gene set enrichment analyses, a right-tailed Fisher's exact test was used to calculate a pathway p-value determining the probability that each biological function assigned to that data set was due to chance alone. All enrichment scores were calculated in IPA using all transcripts that passed QC as the background data set. Upstream regulator analysis is based on prior knowledge of expected effects between regulators and their known target genes according to the IPA database. The prediction of activation state is based on the global direction of changes of differentially expressed genes; a z-score is calculated and determines whether gene expression changes for known targets of each regulator are correlated with what is expected from the literature for an activation of this pathway.

Enrichment analysis to annotate the function of WGCNA-defined gene modules was conducted using Metascape, a platform used for inclusive gene list annotation and source analysis (https://metascape.org/)[20].

## Predicting PASI scores using Gaussian process and ridge regression models

The end goal of this study was to determine the overall disease endotypes and phenotypes responsible for the progression and severity of psoriasis. Machine learning (ML) models are being increasingly adopted across the life sciences as decision-making tools. ML models involve general-purpose algorithms that learn patterns from high-dimensional datasets for performing prediction tasks. On the other hand, statistical models (e.g., linear regression) are often more suitable for inference (i.e., distinguishing whether one or more variables are signals or noise)[21]. Most machine learning algorithms involve supervised tasks, i.e., mapping one or more feature inputs to corresponding labels (i.e., the ground truth) and making predictions for similar but unseen labels. A neural network is a popular machine learning

choice for supervised tasks but requires thousands of labelled examples for learning. Given the size of this study's dataset (n 139 patients and 339 timepoints) a Gaussian process regression (GRP) was explored[22].

Gaussian processes (GPs), a family of Bayesian models, have been shown to perform well on a range of modelling tasks given a limited amount of data[23–26].

Two key features allow GPs to model a range of problems with limited data. Firstly, in the absence of testing data, GPs provide measures of uncertainty to determine how close predictions are to examples in the training dataset[22]. Secondly, GPs attempt to model a distribution over functions $f(x)$. This is specified using a kernel covariance function that makes some basic prior assumptions about the relationship such as whether the functions are smooth, linear or rough. The kernel covariance function also makes the basic assumption that data inputs that are closely related are more likely to have similar labels. When the relationship is unknown, a popular choice of kernel is the non-linear Matern 5/2 kernel, which was adopted in this study[22]. Further, the kernel function can be decomposed into low-order functions that can be used to model feature inputs additively[27]. Many relationships can be decomposed into additive parts, for instance, the price of a building can be broken down into the individual building materials. If the relationship depends jointly on additive low-order interactions, the sum of kernels can be used to model the relationship. If not, the kernels will still specify a suitable model. In this study, an additive non-linear kernel function and Gaussian process model were implemented using the GPflow python package (version 2.5.2)[28].

During training, the kernel hyperparameters including the length scale $\iota$ and variance $\sigma^2$ were tuned for each feature input by maximising the probability of observing the data points, known as the marginal likelihood, on an independent 10-fold validation dataset. Predictions were then obtained from the trained GPR regression.

As a baseline comparison model, a linear ridge regression model was chosen. This model is a special case of a linear regression model that includes an L2 regularisation function to reduce overfitting. A range of hyperparameters were chosen to tune the L2 regularisation parameter on an independent 10-fold validation dataset.

Predictions from the trained models were assessed using shuffled 10-fold held-out testing patient datasets. Performance metrics such as the coefficient of determination ($R^2$) and mean absolute error (MAE) were calculated for each shuffled dataset.

A total of 4 GPR regression and 4 ridge regression models were trained and tested for predicting PASI scores using several features (i.e. Demographics + clinical features, Skin factors and Skin RNA modules). Notably, to reduce the influence of extreme outliers and allow more direct comparison between feature inputs, models adopted log transformed PASI scores and feature inputs normalised using the robust scaler technique. The robust scaler technique uses the feature median and interquartile range rather than the mean and standard deviation.

We compared the performance of the GPR model to a linear ridge regression model. Through comparison to this baseline method, we were able to determine whether the non-linearity captured by the GPR models improved the predictive accuracy (Supplementary Tables 3–5).

To determine the features driving the model relationships, SHAP (SHapley Additive exPlanations) method[29] was adopted. SHAP method is a popular and model-agnostic approach for explaining machine learning outputs. SHAP assesses the impact of each feature on the predicted output while keeping other features unchanged. Higher SHAP values indicate features causing a higher change in the predicted output value and lower SHAP values indicate the opposite.

## Statistics and reproducibility

Module identification with WGCNA was done using Pearson correlation with pairwise complete observations. Similarly, trait correlations with WGCNA module eigengenes and ICA latent factors were calculated using Pearson correlation with pairwise complete observations. The trait correlations were calculated in lesional skin, non-lesional skin and blood

separately and with different sample subsets for each variable type: correlations with the disease endotype variables (i.e. age of onset, onset type, anti-TNF naive status, PsA, sex, age, BMI, HLA-Cw6 status) were calculated using the baseline samples from both drug cohorts and correlations with PASI were calculated for each drug cohort separately using samples from all time points. To account for multiple testing within each tissue, the p.adjust function with method "fdr" in R was applied to the trait correlation p-values for modules and factors separately. Additionally, correlations of module eigengenes and latent factors with per-sample imputed absolute cell fractions were calculated using Kendall's tau coefficient and FDR-adjusted p-values were derived as above. P-values from gene-level differential expression modelling of disease severity were adjusted using Storey's method[12]. Further details about statistical methodology are available under the relevant subsections in the materials and methods section.

## Results

### Study design

Our study design was previously reported[3,30]. The timing of sample collection and details of sample numbers are illustrated in Fig. 1 (see main and supplementary Materials and Methods). We studied 89 subjects with stable plaque psoriasis initiating biologic therapy, 82 of whom provided skin biopsies (41 with adalimumab and 41 with ustekinumab; 400 total samples) and 83 of whom provided blood samples (40 with adalimumab and 43 with ustekinumab; 318 total samples) (discovery cohort) (supplementary Materials and Methods). We replicated findings in a further cohort of 57 subjects who provided skin samples (29 with adalimumab, 28 with ustekinumab; 276 total samples). Power calculations based on Guo et al.[6] and a pilot study[5] indicated a discovery cohort sample size requirement of 40 (Supplementary Methods and Supplementary Fig. 1). Subject characteristics for included participants are shown in Supplementary Table 1a, b.

### Identification of gene expression signatures in skin and blood

To define the relationship between transcriptional signatures and clinical phenotypes, we used two complementary methods for dimensionality reduction, Weighted Gene Correlation Network Analysis (WGCNA) and Independent Component Analysis (ICA) (Fig. 2a, b, Materials and Methods)[4]. In brief, WGCNA groups genes into modules based on co-expression and ICA identifies latent variables that describe patterns of expression variation in the data[4]. WGCNA-derived eigengenes summarise the main trend in co-expressed gene modules, while ICA-derived metagenes emphasise independent patterns of gene expression[4]. To facilitate comparisons between lesional and non-lesional skin across time, all skin samples from the adalimumab and ustekinumab drug cohorts at weeks 0, 1 and 12 were analysed together. Due to extensive transcriptomic variation between skin and blood, these tissues were analysed separately.

WGCNA identified 34 co-expressed gene modules in lesional skin and nonlesional skin across all time points (Supplementary Fig. 2a). Individual modules contained between 34 and 2677 genes. ICA identified 24 factors in lesional skin and nonlesional skin across all timepoints. WGCNA of the blood RNA-seq data identified 26 co-expressed gene modules (Supplementary Fig. 2b) with individual modules containing between 50 and 1333 genes. ICA identified 21 factors in blood. Many of the factor metagenes were highly correlated with module eigengenes in both tissues (Supplementary Fig. 3a, b), indicating that the methods converged on similar key signatures, providing cross-validation.

To define disease and severity endotypes, module eigengenes and factor metagenes were correlated with clinical phenotypes and *HLA-Cw6* genotype status at baseline, which were designated as disease endotypes; and with PASI across all time points, termed disease severity endotypes. These associations were replicated by independent testing within the replication cohort (Fig. 2a, b).

To further define the functional relevance of the co-expressed WGCNA modules and ICA factors, systems analysis was performed.

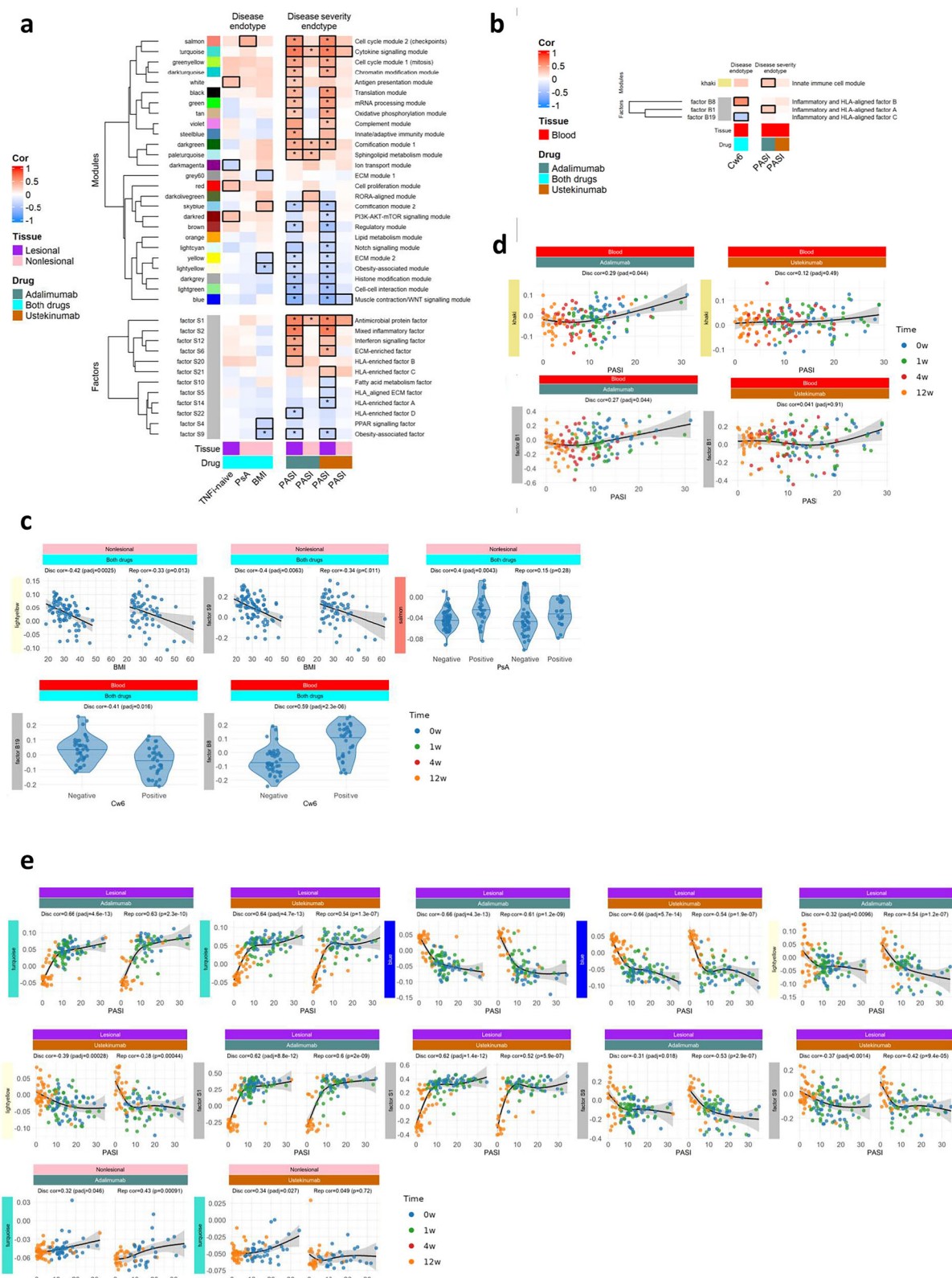

Further information about the modules and factors, including the top pathway enrichments and exemplar aligned genes, is available in Supplementary Data 5 and described in the results below.

We carried out module preservation analysis to assess the degree to which skin modules were preserved in blood and vice versa. The skin modules black (translation) and tan (oxidative phosphorylation) exhibited

strong evidence for preservation in blood and were found to have the most significant gene overlap with the gold (translation) module in blood (Supplementary Fig. 4a, b). The green module (mRNA processing) was also strongly preserved in blood and exhibited significant gene overlap with the olivedrab (oxidative phosphorylation) module in blood (Supplementary Fig. 4a, b).

**Fig. 2 | Systems-level gene modules and latent factors correlate with disease and disease severity endotypes in psoriasis.** Heatmaps display pearson correlations between (upper panels) WGCNA module eigengenes or (lower panels) ICA latent factors and clinical/demographic variables for (**a**) skin and (**b**) blood in the discovery cohort. Benjamini–Hochberg multiple testing correction was done separately for modules and factors and skin and blood. Black outlines denote correlations significant at false-discovery rate (FDR) ≤ 0.05; asterisks indicate the correlation replicates (same sign, P < 0.05) in the independent replication cohort. Only modules/factors with at least one replicated correlation are shown; descriptors summarise the dominant biological theme of each module/factor. Panels (**c**–**e**) illustrate three examples: Baseline clinical endotypes—Pearson correlations with BMI in skin and *HLA-C\*06:02* genotype in blood (**c**). Disease severity endotypes—non-linear relationships between eigengenes and PASI across all visits in blood (**d**) and skin (**e**). Significant disease severity associations were first identified by pearson correlation and then visualised with spline curves. Points are coloured by visit (blue = week 0, green = week 1, red = week 4, orange = week 12). Violin plots include a median line. Curves represent natural-spline fits (3 d.f.) with 95% confidence bands, as in Fig. 1 plus. HLA human-leucocyte antigen, FDR false-discovery rate.

## Disease endotype

Significant associations of clinical phenotypes with WGCNA modules and ICA factors were found in both lesional skin and nonlesional skin (Fig. 2a) at baseline. Notably, we observed that the lightyellow module (insulin and hormone secretion) and factor S9 (obesity-associated) each displayed significant and replicable negative associations between their eigengene expression and i) BMI in nonlesional skin and ii) PASI in lesional skin (Fig. 2a, c; Supplementary Fig. 5). This supports our earlier observed association between high BMI and poor clinical response in a larger cohort of patients[31]. Functional enrichments for lightyellow and factor S9 included: secretory pathways, hormone signalling, transport of small molecules and Wnt signalling (Supplementary Data 5). Twenty-one genes within the intersection of lightyellow and factor S9 were independently negatively associated with BMI (Fig. 3a, b). Notably 14 of these genes, including *SCGB1D2* (Secretoglobin Family 1D Member 2), *MMP7*, *DNER* (Delta/Notch Like EGF Repeat, regulates adipogenesis) and *PDE9A* (stimulates lipogenesis) were negatively associated with PASI in lesional skin across all time points (Fig. 3c). Deconvolution of bulk RNA-seq data with single cell RNA-seq data from 38,274 skin cells[18] revealed that lightyellow, factor S9, and the 14 gene core BMI/PASI signature (including *SCGB1D2, DNER* and *PDE9A*) were highly enriched with genes expressed within the pilosebaceous unit (Supplementary Figs. 6, 7b, 8).

Two blood factors were significantly associated with HLA-Cw6 genotype status (as a binary trait), with factor B8 (inflammatory and HLA-aligned B) being positively correlated and factor B19 (inflammatory and HLA-aligned C) negatively correlated (Fig. 2b). These factors were enriched for antigen processing, graft versus host disease and allograft rejection, with these enrichments being driven predominantly by *HLA*- genes (Supplementary Data 5). The factors were highly aligned with the expression of genes in the 6p21 gene region, including *PPP1R18, HLA-DRB1, HSPA1L, GPSM3, HSPA1A, LY6G5B, CSNK2B* and *TUBB* (factor B8); and *HLA-DRA, HLA-DQB1, AGPAT1, PSMB9, HLA-DQA1, HLA-DRA* and *HLA-DMB* (factor B19).

## Disease severity endotype

Few previous studies have studied the relationship between gene expression patterns and disease severity across different tissues in psoriasis[32,33]. We therefore investigated whether gene expression within lesional skin, non-lesional skin and blood was associated with whole body disease severity scores at the time of sampling. PASI is the gold-standard disease severity measure and represents the average redness, thickness, and scaliness of psoriasis lesions weighted by the area of involvement (Fig. 1b, top panel)[34].

**Lesional skin.** Remarkably, 21 out of 34 WGCNA modules in lesional skin showed highly significant and reproducible correlations with disease severity in at least one of the drug cohorts (Fig. 2a); 12 showed positive correlations and 9 negative correlations with disease severity. Cross-correlation of the key modules and factors identified two distinct blocks (i and ii) (Supplementary Fig. 3a). Strikingly, the first block (i) comprised modules and factors (e.g. turquoise module and factor S1; cytokine and anti-microbial peptide signalling) positively associated with disease severity whereas the second block (e.g. yellow module and factor S9; ECM and Insulin and hormone secretion signalling) (ii) was negatively associated with disease severity (Fig. 2a; Supplementary Fig. 3a). Exemplar scatter plots are shown in Fig. 2c and Supplementary Fig. 9.

To gain further insight, we deconvoluted the bulk RNA-seq data using single-cell RNA-seq data[18]. Cluster analysis showed the presence of two main blocks that were distributed according to positive or negative disease severity associations (Supplementary Fig. 6). For example, we observed resolution of keratinocyte subsets into two distinct clusters, consistent with previous single cell[35] and spatial transcriptomic studies[36]. Thus, keratinocyte-1 (KRT1+, KRT10+ S100A8/9+ spinous), keratinocyte-4 (KRT1+, KRT10+ S100A8/9+ spinous), keratinocyte-6 (KRT5+, KRT14+ KRT1+, KRT10+ PCNA+ supra-basal) and keratinocyte-7 (KRT5+, KRT14+, COL17A1+ basal) subsets (Supplementary Fig. 7a) correlated with positively-associated disease severity modules and factors (e.g. turquoise module, darkgreen module and factor S1) whereas keratinocyte-3 (KRT5+, KRT14+, KRT1+, KRT10+ supra-basal), keratinocyte-5 (KRT1+, KRT10+ IVL+ supra-spinous) and keratinocyte-8 (KRT5+, KRT14+, COL17A1+, PCNA+ basal)[35] subsets (Supplementary Fig. 7a) aligned with negatively-associated disease severity modules and factors (e.g. yellow module, skyblue module and factor S9) (Supplementary Fig. 6). Notably, each cluster comprised basal, suprabasal and spinous subsets, reinforcing a mechanistic model of a switch in keratinocyte differentiation program and phenotype with disease severity progression[18,35]. Additionally, and in line with the current pathophysiological understanding of the role of innate and acquired immunity in psoriasis[35,37], positively-associated disease severity modules and factors showed strong associations with myeloid-1 (dendritic cells/macrophages), T cell-2 (Th1, Th17), T cell-3 (cytotoxic T lymphocytes)and venule-2 cells (which regulate the trafficking of immune cells into tissues) (Supplementary Figs. 6, 7e–g). Deconvolution of negatively-associated disease severity modules and factors revealed enrichment for fibroblast-5 (SFRP+, MFAP5+; f2/3 stromal/mesenchymal)[38] and Langerhans cells (Supplementary Figs. 6, 7b–d).

**Nonlesional skin.** Although clinically resembling normal skin, it is recognised from gene and protein expression studies that nonlesional skin represents a pre-psoriatic state that is primed to develop into psoriasis[39]. We extend these findings to show that in nonlesional skin, eigengene expression of three WGCNA modules (turquoise, darkgreen (cornification 1) and paleturquoise (sphingolipid metabolism) and factor S1 significantly and reproducibly positively correlated with whole body disease severity in the adalimumab group (Fig. 2a, c). Of note, there were no significant negatively associated disease severity modules or factors in nonlesional skin.

**Blood.** Limited previous studies have systematically investigated mRNA biomarkers of psoriasis disease severity in blood[40]. The khaki (innate immune cell; neutrophil degranulation) module and factor B1 (inflammatory and HLA aligned A; phagosome) were positively associated with disease severity in the adalimumab cohort, and deconvolution indicated enhanced representation of neutrophils (Fig. 2b, Supplementary Fig. 10, Supplementary Data 5). Strikingly, in the ustekinumab cohort there were no correlations reaching statistical significance (Fig. 2b, c).

## Predicting disease severity using machine learning

To predict disease severity from modules and factors, we employed additive *Gaussian*-process regression (GPR; Supplementary material and methods). GPR was well-suited to our "small-n, large-p" design, which comprised

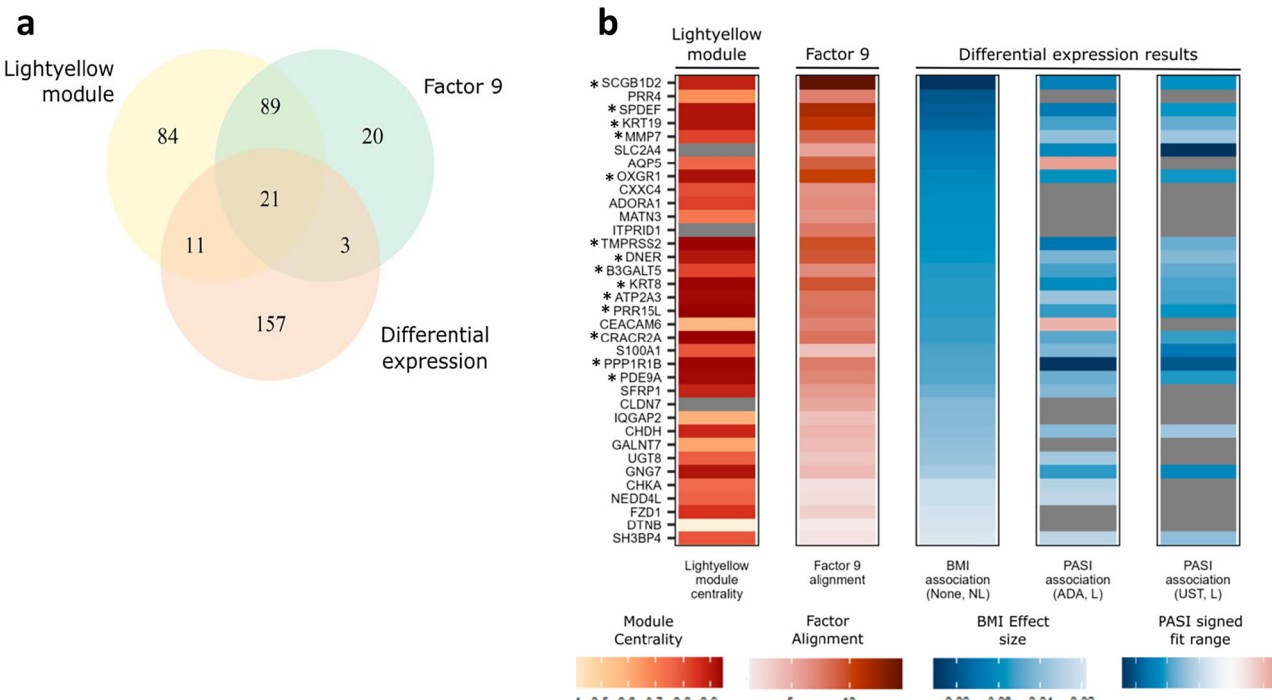

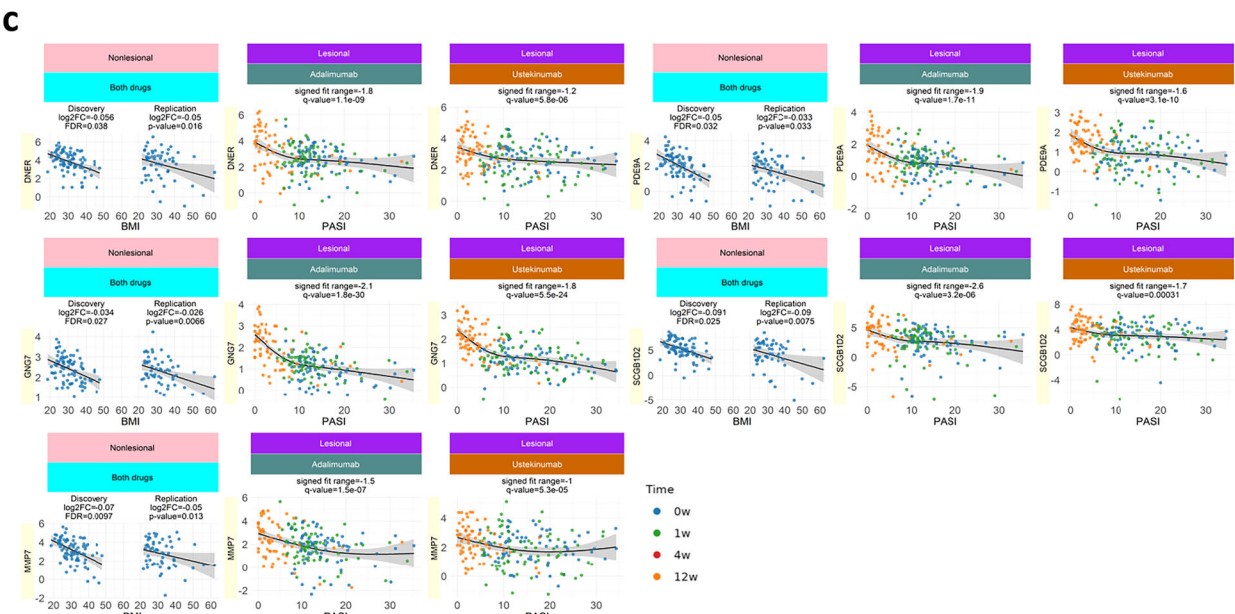

**Fig. 3 | A BMI-related transcriptomic signature in non-lesional skin is also linked to disease severity in lesional skin. a** Venn diagram showing overlap between genes that (i) belong to the lightyellow WGCNA module, (ii) have high alignment to ICA factor 9 (absolute scaled loading > 5), and (iii) are differentially expressed with BMI in nonlesional skin (Benjamini-Hochberg FDR < 0.05 after multiple testing correction, *limma*). **b** Heatmap of key genes within this shared signature. Columns correspond to genes; rows encode three attributes: membership of the light-yellow module (yellow), strength of loading on factor 9 (red intensity), and significance of BMI association (blue intensity). Genes marked with * are also significantly associated with PASI in lesional skin (Storey's q < 0.05 after multiple testing correction)

in both biologic cohorts. **c** On the left, the association between expression (log CPM) and BMI in non-lesional skin is plotted; best fit lines with 95% confidence intervals for the predicted values from expression-BMI regression are shown, which illustrate consistent direction of effect. On the right, the association between expression and PASI in lesional skin is plotted; curves represent natural-spline fits (3 d.f.) with 95% confidence intervals for the predicted values and signed fit range is shown in the plot title, which is equal to the minimum to maximum range of the fitted expression, with the sign given by the sign of the gradient at PASI = 0. BMI body-mass index, CPM counts per million, ADA adalimumab, UST ustekinumab, LS lesional skin, NLS non-lesional skin.

718 samples from 146 subjects and approximately 20,000 transcripts within 60 modules and 45 factors across skin and blood, as this method combines (i) non-linear flexibility with (ii) a Bayesian framework that returns per-sample credible intervals, an essential read-out for clinical

risk-stratification. Moreover, its additive kernel decomposition facilitated transparent attribution of individual module and factor contributions via SHAP values (Fig. 4b, d), which cannot be obtained as cleanly from tree-based ensembles or neural networks.

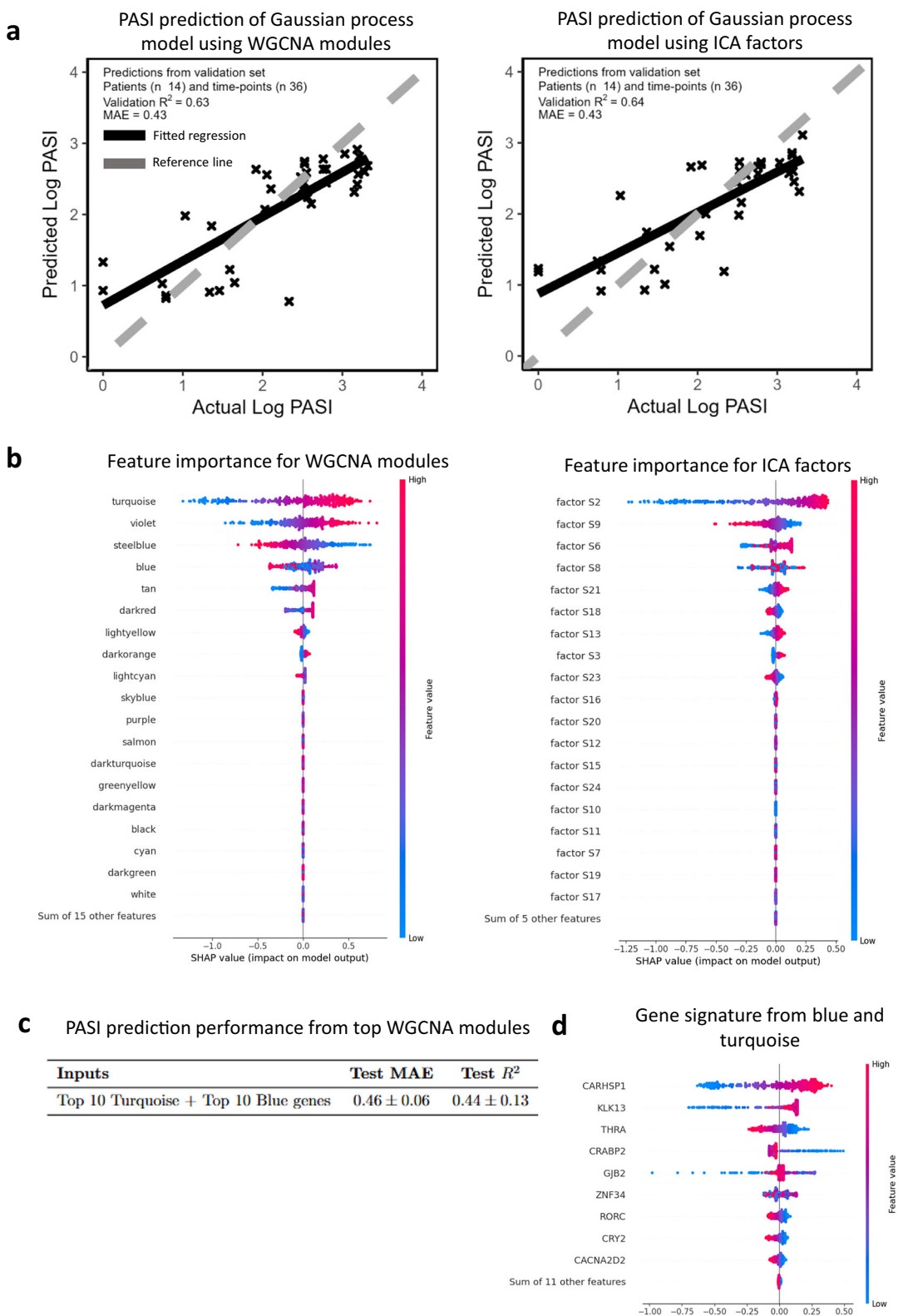

**Fig. 4 | Gaussian process regression accurately predicts log PASI from transcriptomic modules and factors.** Data from 139 individuals (discovery + replication) were randomly partitioned into training (80%, $n = 111$), validation (10%, $n = 14$) and test (10%, $n = 14$) sets. **a** Predicted versus observed log PASI for the optimal models built from WGCNA eigengenes (left) or ICA latent factors (right) on the validation set. **b** SHAP summary plots rank feature importance. Positive SHAP values (right of 0) indicate features pushing predictions up (higher PASI); negative values (left) push predictions down. **c** Removing all but the top eigengenes/factors shows that a parsimonious 10-gene signature (derived from blue + turquoise modules) preserves predictive performance. **d** Applying SHAP to the top performing modules (blue and turquoise) identified a 9 gene signature predictive of log PASI. Comparison with a baseline linear ridge-regression (see Supplementary Fig. 11) highlights the gain from modelling non-linear interactions, as in Fig. 1; SHAP SHapley Additive exPlanations.

## Table 1 | PASI prediction performance of additive Gaussian process models

| Inputs | Test MAE | Test $R^2$ |
|---|---|---|
| Demographics, clinical features, WGCNA modules and ICA factors | 0.47 ± 0.07 | 0.49 ± 0.10 |
| All WGCNA modules | 0.45 ± 0.07 | 0.53 ± 0.09 |
| Turquoise module | 0.45 ± 0.07 | 0.52 ± 0.10 |
| Violet module | 0.61 ± 0.09 | 0.12 ± 0.13 |
| Steelblue module | 0.61 ± 0.10 | 0.14 ± 0.15 |
| Blue module | 0.45 ± 0.07 | 0.54 ± 0.09 |
| Tan module | 0.55 ± 0.06 | 0.29 ± 0.09 |
| All ICA factors | 0.46 ± 0.06 | 0.51 ± 0.09 |
| Factor S1 | 0.45 ± 0.06 | 0.51 ± 0.09 |
| Factor S2 | 0.46 ± 0.08 | 0.51 ± 0.11 |
| Factor S6 | 0.52 ± 0.07 | 0.35 ± 0.14 |
| Factor S9 | 0.58 ± 0.09 | 0.21 ± 0.10 |
| Factor S8 | 0.66 ± 0.09 | 0.00 ± 0.04 |
| Factor S21 | 0.66 ± 0.09 | 0.00 ± 0.03 |

Shown are means ± SD for 20 randomly shuffled 10-fold testing datasets.

A linear ridge-regression (RR) baseline was trained for comparison. In five-fold cross-validation GPR achieved MAE = 0.45 ± 0.07 and $R^2$ = 0.53 ± 0.09, outperforming RR (ΔMAE = +0.06; Δ$R^2$ = −0.08).

The final GPR and RR models, fitted to the combined Discovery and Replication cohorts, tracked the training data well ($R^2$ = 0.59–0.78; Table 1, Supplementary Table 3). Crucially, only GPR supplies calibrated uncertainty bands around each prediction, complementing global metrics such as MAE and $R^2$.

To assess the GPR and RR models on unseen data points, models maximised on an independent validation set were retrained and assessed on additional held-out subject-level inputs (Table 1). This suggested demographics and clinical features related poorly to disease severity (Supplementary Fig. 11), while both RNA eigengenes and skin factors strongly related to disease severity in both the validation and testing datasets (Fig. 4a and Table 1). To demonstrate how our models can make clinically meaningful predictions, a random selection of subject predictions were plotted over time (Supplementary Figs. 13, 14).

To assess feature importance, we used SHAP (SHapley Additive exPlanations) which shows the contribution of each skin RNA module and skin factor for predicting disease severity (Fig. 4b)[29].

Despite differences in model methodology (i.e. linear vs. nonlinear), both regression techniques prioritised similar eigengenes and skin factors, although GPR was more selective than RR (Fig. 4b and Supplementary Fig. 12b). Overall, both methods highlighted turquoise, darkred (PI3K-AKT-mTOR signalling), steelblue (innate/adaptive immunity), violet (complement) and tan (oxidative phosphorylation) modules as important contributors to disease severity prediction (Fig. 4b and Supplementary Fig. 12b). Factors S1 and S2 (mixed inflammatory), which are inter-correlated (Supplementary Fig. 3a), were identified by SHAP analysis of RR and GPR models respectively as the most influential skin factors for predicting PASI (Fig. 4b). Additionally, factors S9 and S6 (ECM signalling) appeared as top features influencing disease severity (Fig. 4b and Supplementary Fig. 12b). By analysing both MAE and $R^2$ values, we identified the turquoise module as the strongest positive and the blue module as the strongest negative contributors to disease severity prediction (Table 1). In order to gain a deeper understanding of key genes driving disease severity prediction, the GPR models were retrained with only the top 10 aligned genes from selected modules and factors correlating with disease severity (Fig. 4c and Supplementary Tables 3, 4). A SHAP analysis of the top 10 aligned genes from turquoise and from

blue identified CARHSP1, KLK13, CRABP2, and GJB2 (turquoise, cytokine) and THRA, ZNF34, RORC, CRY2 and CACNA2D2 (blue, WNT signalling) as the 9 key genes driving disease severity prediction (Figs. 4d, 6d).

Factors S18 (HLA-DQA1*01/HLA-DRB1*15-associated) and S21 (HLA-enriched factor C) were among the top predictors of disease severity according to both models (Fig. 4b and Supplementary Fig. 12b). Most of the genes highly aligned with these factors were HLA-encoding, with HLA-DQB1, HLA-DQA1, HLA-DRA, HLA-DRB1 and HLA-DRB5 aligning with factor S18, and HLA-E, HLA-DQB2, HLA-DQA2 and GSTM1 aligning with factor S21 (Fig. 5A). Previous studies have reported associations between HLA genotypes and psoriasis severity[41]. We therefore investigated associations between HLA genotypes and the expression of genes and factors. Among the strongest genotype-gene correlations were those between HLA-DQA1*01 genotype and the expression of HLA-DQA1 and HLA-DQB1, and between HLA-DRB1*15 and expression of HLA-DRB5 and HLA-DRB1 (Fig. 5B, C). Additionally, the two strongest genotype-factor relationships were between HLA-DQA1*01 and HLA-DRB1*15 genotypes and factor S18 (Fig. 5B). We observed three clusters of patients corresponding to factor S18, its constituent genes and the HLA genes. Individuals with both the DRB1*15 and DQA1*01 genotypes had high factor S18 expression and high expression of HLA-DQB1, -DQA1, -DRB1 and -DRB5; subjects with the DQA1*01 genotype only had moderate factor S18 expression and high expression of HLA-DQB1 and -DQA1, while those with neither allele had low factor S18 expression and low expression of all four of the aforementioned genes (Fig. 5A).

Given that these HLA-aligned factors were not associated with disease severity across timepoints (Fig. 2a), we investigated why they were considered important for prediction (Fig. 4c). There was no apparent clustering of samples by tissue type (Fig. 5A) and the expression of these factors displayed strong correlations between paired lesional and nonlesional samples (Fig. 5D), suggesting that the expression of these factors does not depend on tissue type. We next considered associations between the factors and baseline disease severity. Factors S1, S2, S17 (matrisome and cytokine receptor) and S18 were significantly correlated (FDR < 0.05) with baseline PASI in lesional skin; factor S18 was the most significantly associated (Pearson's $r = -0.28$, adjusted $p$-value = 0.01) (Supplementary Table 5). Furthermore, the patient clusters based on the HLA-DQA1*01 and HLA-DRB1*15 genotypes were significantly associated with baseline PASI ($p$-value = 0.007) (Fig. 5E, F). These data indicate a link between these genotypes, expression of HLA-DQB1, -DQA1, -DRB1 and -DRB5 and psoriasis severity at baseline.

### Disease severity endotype gene level analysis

**Lesional skin.** Extending our investigation to gene level (Fig. 6a), we found a significant nonlinear association between expression of 4108 genes in lesional skin and psoriasis disease severity (Fig. 6b, c). The majority (2926 [71.2%]) of these genes showed an association independent of biologic treatment modality. Approximately 40% of disease severity-associated genes showed an overall positive relationship and approximately 60% a negative relationship with disease severity (Fig. 6c). Previous studies have paid less attention to genes negatively correlated with PASI, but these may represent important regulatory pathways constraining inflammatory responses and may represent novel therapeutic targets.

Of the genes which were associated with disease severity for both drugs, 33% (968 genes) were assigned to the turquoise module (cytokines) and almost 25% (698 genes) were assigned to the blue module (WNT), which were the top-ranked modules in the GPR model positively and negatively associated with disease severity, respectively (Figs. 4b, 6c). Genes highly positively correlated with disease severity within the turquoise module are shown in Fig. 6d and Supplementary Fig. 15. Interestingly, the majority of the GPR-derived core gene signature driving disease severity prediction (highlighted in Fig. 6c–e) were not previously recognised to be psoriasis disease severity-associated. Network analyses identified S100 family

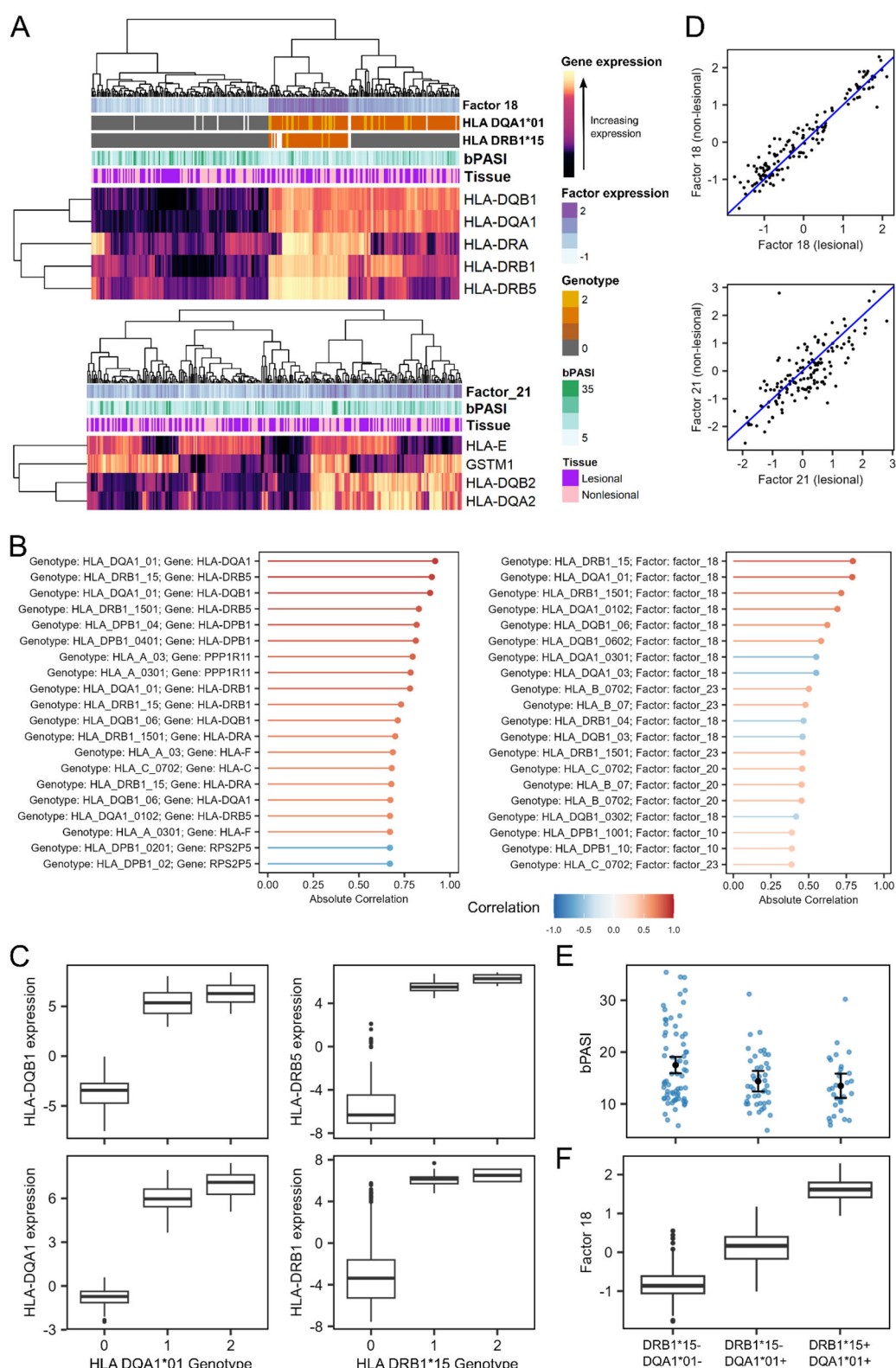

**Fig. 5 | Latent factor S18 defines HLA-driven patient clusters associated with baseline disease severity.** ICA factor S18, derived from combined lesional and nonlesional skin data, segregates patients into clusters differing in class-II HLA genotype and baseline PASI. **A** Two heatmaps for week-0 samples. Upper: expression of factor S18 and its highly aligned genes. Lower: expression of factor S21 (another HLA-related factor). Tracks show tissue type (lesional/nonlesional) and baseline PASI (bPASI). For S18, additional tracks display *HLA-DQA1*01* and *HLA-DRB1*15* genotype dosage (0/1/2; grey, orange, yellow respectively, white is missing). **B** Bar plots of top genotype-gene (left) and genotype-factor (right) correlations

in lesional skin. **C** Boxplots of HLA gene expression stratified by genotype which show first quartile, median, third quartile, minimum and maximum values, and points to indicate outliers. **D** Scatter plot comparing S18 expression in matched lesional vs nonlesional skin; the identity line (blue) marks equal expression. **E** Estimated mean bPASI (95% CI) by genotype from linear modelling. **F** Distribution of S18 expression by genotype in lesional skin. The analysis in this figure was carried out with *n* = 139 biologically independent subjects. HLA human-leucocyte antigen, bPASI baseline PASI, LS lesional skin.

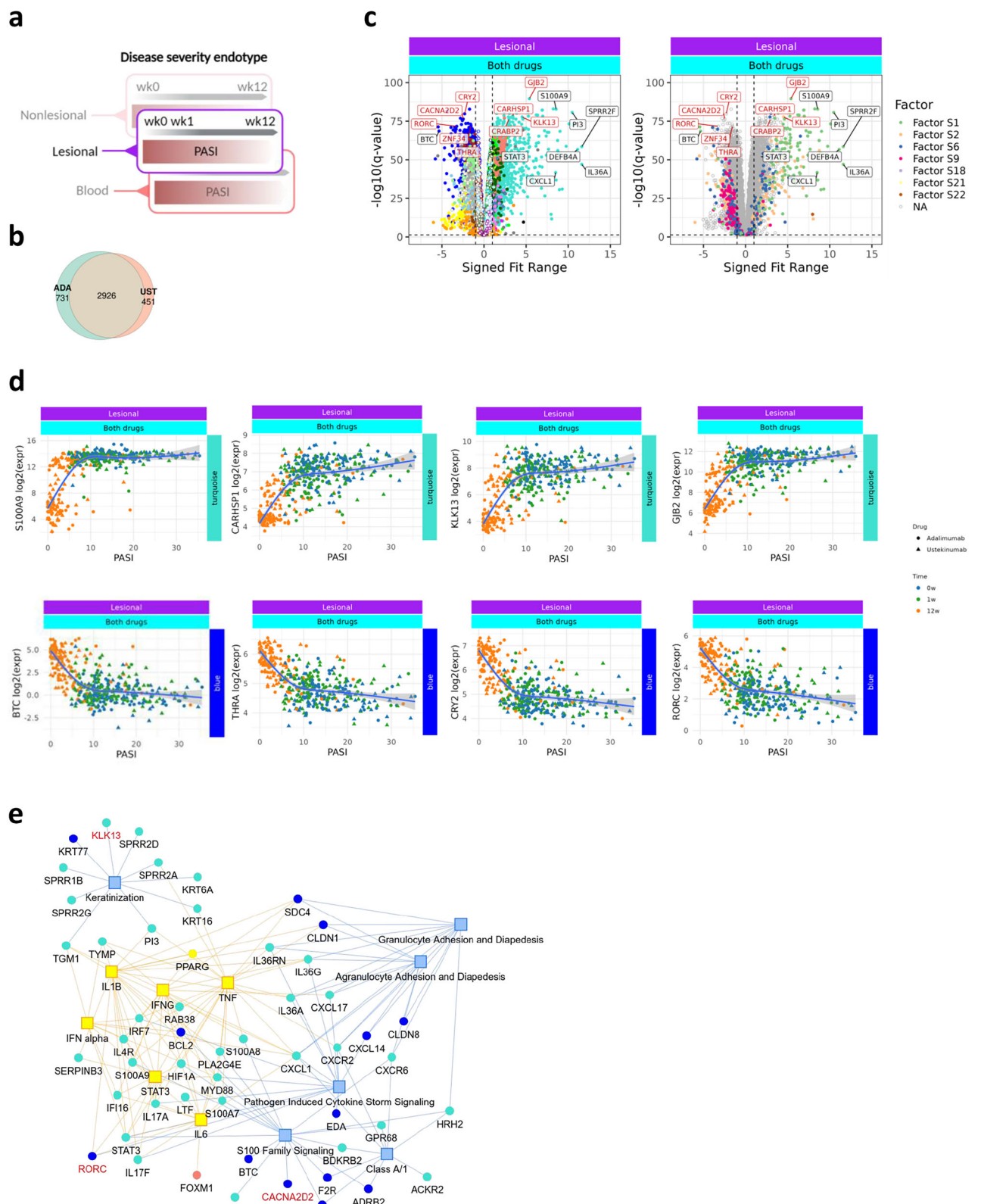

**Fig. 6 | Disease severity endotypes in lesional skin are largely independent of biologic drug. a** Workflow: non-linear (natural-spline 3 d.f.) regression was used to relate gene expression to PASI separately in adalimumab and ustekinumab cohorts; significant genes ($q < 0.05$) were combined for downstream analysis. **b** Venn diagram summarising the overlap of significant genes between treatment arms. **c** Volcano plot for the pooled dataset showing –log10 $q$ versus signed fit range (difference between maximum and minimum fitted expression across the PASI span; sign reflects direction). Left panel colours genes by WGCNA module, right by ICA factor; exemplar genes are annotated. **d** Example turquoise- and blue-module genes plotted as log2-CPM versus PASI with spline fits. Curves represent natural-spline fits (3 d.f.) with 95% confidence intervals for the predicted values. **e** Network view (Ingenuity Pathway Analysis): circles = genes (module-coloured), yellow squares = predicted upstream regulators, blue squares = enriched canonical pathways. ADA adalimumab, UST ustekinumab; others as before.

signalling, keratinisation, interferon, chemokine, granulocyte diapedesis and matrisome signalling as key canonical pathways (Fig. 6e, Supplementary Fig. 16). WNT and collagen genes were prominent within the negatively-associated disease severity network (Supplementary Fig. 17). Network analyses identified TNF, IFN-gamma, STAT3, IL-1B and IL6 as key upstream regulators of disease severity-associated genes (Fig. 6e, Supplementary Fig. 16). 448 disease severity-associated genes were identified as being highly aligned with a factor. Half of these were turquoise module genes (246 [50.4%]), of which the majority positively aligned with factor S1 or factor S2, underscoring the similarity between these factors. Factor S9 genes were highly correlated with light yellow (obesity, 103 genes), as reported above.

Additionally, in keeping with their correlation with single cell keratinocyte signatures (Supplementary Fig. 6), darkgreen (positively associated with PASI) and skyblue (negatively associated with PASI) contained cornification genes, including ABCA12, SBSN, KLK7, and PCSK6 in darkgreen, and FLG, LOR, GAN, and LCE5A in skyblue; the association of these genes with PASI is illustrated in Supplementary Fig. 15d. This highlights that different sets of skin barrier genes exhibit distinct patterns of co-expression and associations with disease severity, with some being positively associated with PASI and others negatively associated.

A small proportion (29%; Fig. 6b) of the disease severity-associated genes in lesional skin were drug-specific and these typically represented subtle differences between the drugs (Supplementary Fig. 15b).

**Nonlesional skin**. Next, we investigated disease severity-associated genes in nonlesional skin. Unsurprisingly, we found that the numbers of disease severity-associated genes in nonlesional skin were much lower than in lesional skin (145 genes; Supplementary Fig. 18a); similarly, the -log10 p-values and signed fit ranges were lower than those in lesional skin. Several disease severity-associated genes in lesional psoriasis, such as S100A7 and PI3, were also found in nonlesional psoriasis, whereas the cytokine genes IFN-gamma, IL-17, IL-36 and CXCL1 were notably absent (Supplementary Fig. 18a). Consistent with findings at the modular level and in contrast to lesional skin, 79.3% of genes in nonlesional skin were biologic-agent specific. Thus, in nonlesional skin, 105 disease severity-associated genes were adalimumab-specific, 19 were ustekinumab-specific and 21 were common to both drugs (Supplementary Fig. 18a). Adalimumab disease severity-associated genes were enriched for interferon signalling, TNF signalling, keratinisation and granulocyte diapedesis pathways, similar to lesional skin, but not the matrisome canonical pathway, whilst ustekinumab disease-associated genes were enriched for keratinisation and antimicrobial peptides (Supplementary Fig. 18b, Supplementary Fig. 19).

**Blood**. As with nonlesional skin, we found that the numbers of disease severity-associated genes in blood were much lower than in lesional skin (181 genes) with lower −log10 q values and signed fit ranges (Fig. 7b, c). Notably, and in line with our module analysis, 100% of the disease severity-associated genes in blood were specific to adalimumab (Fig. 7b, c) and systems analysis showed enrichment of genes related to neutrophil degranulation with gene membership to the navy (innate/adaptive immunity 1), plum (innate/adaptive immunity 2), khaki (innate immune cell) and thistle (inflammatory cell death) modules, in line with our modular analysis (Fig. 7d, e, Supplementary Fig. 20). The adalimumab-specific genes with the greatest disease severity association included the navy module genes ADGRG3, VNN1 and PADI4 and the thistle module gene S100P (Supplementary Fig. 15). Overall, these data highlight an important relationship between expression of neutrophil degranulation genes in blood and psoriasis disease severity in the skin in patients treated with adalimumab.

**PSORT data analysis portal**
To facilitate the interrogation of this skin and blood RNA-seq dataset, we developed a R Shiny web interface[42] (https://shiny-whri-c4tb.hpc.qmul.ac. uk/psort) that facilitates visualisation and exploration of the data. The web interface includes an interactive interface for comparing genes or gene modules of interest in lesional and nonlesional skin against clinical measures, and searchable tables of differentially expressed genes and module comparisons.

## Discussion
Here, we integrated clinical, genomic and transcriptomic data derived from the skin and blood of psoriasis patients at baseline and across early time points during therapy with two classes of biologics. This analysis identified reproducible endotypes (subgroups) of disease associated with clinical phenotypes, disease severity and class of drug treatment. We found that endotypes were most accurately defined by core networks within bulk RNA-seq data that represent direct and indirect interactions of genes across several cell types within tissue compartments. The disease severity endotypes in lesional skin were characterised by a core network of co-ordinated genes and pathways that were largely independent of drug treatment. Positive signatures of disease severity linked innate and acquired immunity whilst negative signatures of disease severity were characterised by matrisome and Wnt signalling pathways. The identification of coordinated gene networks that are negatively associated with psoriasis disease severity points towards regulatory and restraining functions of the interconnected networks in skin which require further study and potentially represent novel therapeutic targets. ICA, but not WGCNA, delineated gene signatures in blood that characterised the HLA-Cw6-associated phenotype of psoriasis. In contrast, disease severity endotypes in nonlesional skin and blood were more specifically connected to the individual biologics. In blood, a key adalimumab disease severity-associated module (khaki, innate immune module) was functionally enriched in neutrophils and genes related to neutrophil degranulation, in line with the concept that the blood transcriptional signature of psoriasis is neutrophil driven[40]. Taken together, our findings support a model in which adalimumab and ustekinumab exert differential effects across tissue compartments with adalimumab exerting greater effects in nonlesional skin and blood, in line with the pleiotropic effects of TNF, where ustekinumab effects appear more specifically targeted to the diseased tissue (lesional skin). These data are also consistent with reduced drug survival due to side effects[43] and increased serious infection risk with adalimumab compared to ustekinumab[44].

Our study aimed to investigate endotypes in psoriasis, a complex task that requires sophisticated and multi-dimensional analytical models. Single-gene differential expression analysis has been widely used to explore the pathophysiological mechanisms of psoriasis, but potentially oversimplifies the complexity of gene interactions and regulation in biological systems, and may be confounded by measures of disease activity, such as PASI, which are unlikely to exhibit simple linear relationships. This may help to explain the limitations of other transcriptomic studies to date; to our knowledge, these have failed to independently replicate response biomarkers in psoriasis.

Applying WGCNA and ICA to the large number of samples in PSORT, we have delineated the direct and indirect interaction of genes across different cell types that potentially drive diverse phenotypes within a clinically homogeneous archetypal immune-mediated inflammatory disease (IMID) - psoriasis. Finding correlations between these two analyses can provide a more comprehensive understanding of gene interactions, as it suggests that these genes not only share expression patterns but also contribute to distinct biological signals or functions.

The coordinated expression between WGCNA modules and ICA factors was reflected in the two main clusters following correlation analysis (Supplementary Fig. 3a). The functional relevance of these two clusters is underscored by their anti-correlation with disease severity (PASI). Thus, WGCNA and ICA identified a first block of modules and factors positively associated with disease severity, which were also prioritised by machine learning methods, and represented cytokine signalling, oxidative phosphorylation, antimicrobial proteins and the extracellular matrix pathways (Fig. 4b). Conversely the second block was negatively correlated with disease

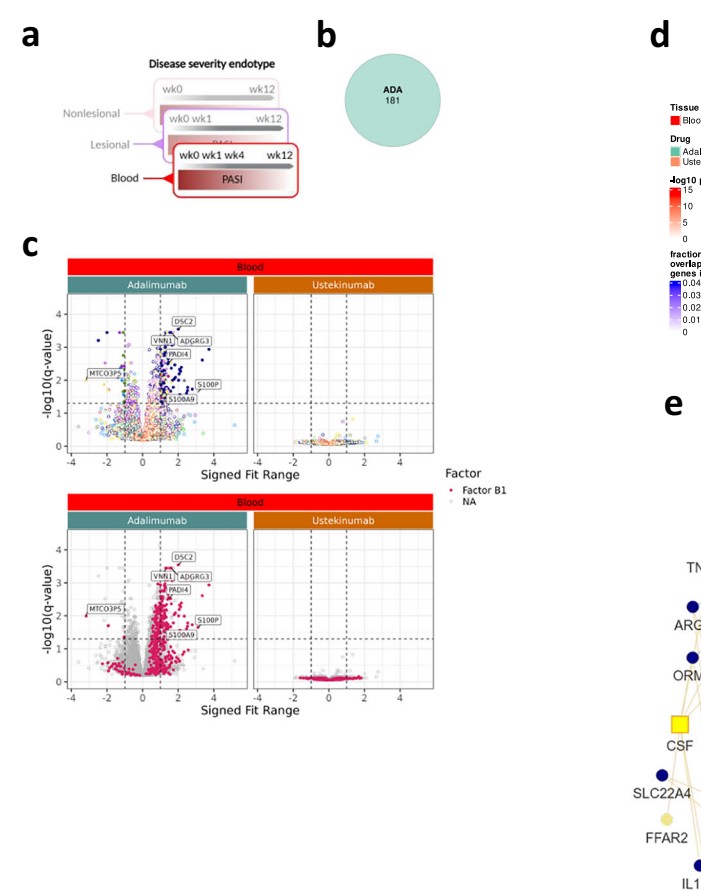

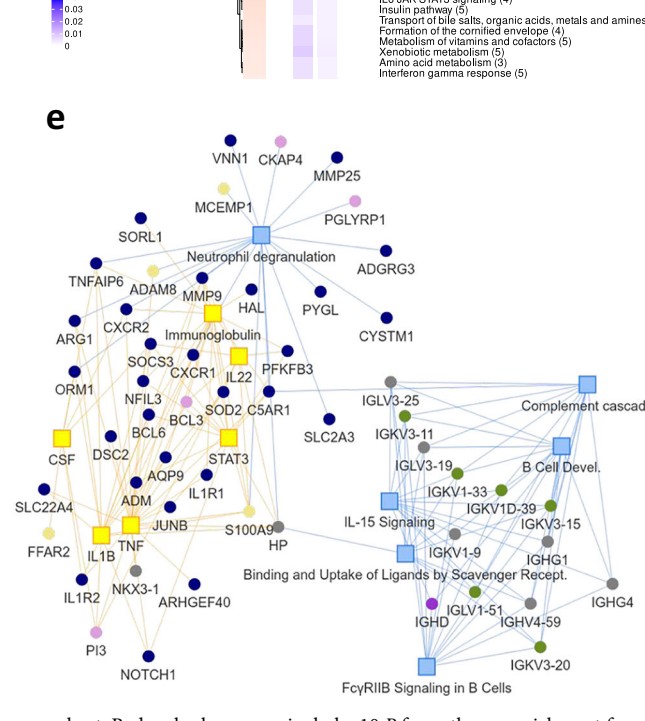

**Fig. 7 | Disease-severity signatures in blood are specific to adalimumab.**
**a** Workflow schematic parallel to Fig. 6. **b** Venn diagram showing no overlap of PASI-associated genes between blood of adalimumab and ustekinumab cohorts ($q < 0.05$). **c** Volcano plots for each drug: –log10 $q$ versus signed fit range, coloured by WGCNA module (upper row) or ICA factor (lower row); illustrative genes labelled. **d** Metascape enrichment heatmap of co-expressed PASI-associated genes in each cohort. Red scale shows nominal –log10 $P$ for pathway enrichment from the hypergeometric test; blue scale shows gene-set overlap fraction after module-size normalisation. **e** Network of adalimumab-specific PASI genes: nodes coloured by WGCNA module; yellow squares mark predicted upstream regulators and blue squares canonical pathways (Ingenuity Pathway Analysis). as before; q, Benjamini–Hochberg FDR-corrected $P$ value.

severity, representing Wnt signalling, insulin and hormone secretion and obesity-associated pathways (Fig. 4b).

Gene expression in skin has been previously linked to obesity[45], with some of these genes associated with psoriasis. Here, we define a 14-gene signature that negatively correlates with BMI within nonlesional skin and psoriasis disease severity within lesional skin, underscoring common pathways and mechanisms. Psoriasis is strongly associated with the metabolic syndrome, obesity and high BMI[46,47], supported by our previous study of 2889 biologic-treated patients[31]. Several of these core BMI/PASI signature genes are candidates for further mechanistic studies. For example, *DNER* (Delta/Notch Like EGF Repeat) regulates adipogenesis by modulating mesenchymal stem cell proliferation[48] and polymorphisms of *DNER* associate with the risk of type II Diabetes in certain populations[49]. In line with our data, *DNER* mRNA was upregulated in lesional psoriasis 12 weeks into therapy with fumaric acid esters[50].

The enrichment of the insulin and hormone secretion signature in sebocytes and the hair follicles following deconvolution is intriguing, given the enhancement of Wnt signalling pathways in our systems analysis.*SCGB1D2* (Secretoglobin Family 1D Member 2), expressed by sebocytes and sweat glands (Supplementary Fig. 8)[18], and previously reported to be down-regulated in psoriasis[51], was one of the top 5 down-regulated genes in white adipose tissue comparing obese insulin-resistant to normal glucose tolerance individuals[52]. Furthermore, a recent spatial transcriptomic study

identified cross-talk between cutaneous inflammation and lipid metabolism in immunocompetent sebaceous glands[53]. *PDE9* regulates energy metabolism and inhibition of phosphodiesterase type 9 reduces obesity and stimulates lipogenesis via PPARs[54]. Interestingly, a PPARg-expressing Treg cell population, identified in a mouse model of psoriasis which suppresses IL17 expression, was lost in obese mice[55]. Furthermore, previous bulk RNA-seq and spatial transcriptomic studies of psoriatic skin identified differentially expressed genes within sebaceous glands that were regulated by PPAR-g, the target of the diabetes drug, pioglitazone, which has shown efficacy in psoriasis[56]; we observe *PPARGC1A* within the light-yellow module, in line with these findings. The intersection of psoriasis, metabolic syndrome, and type 1 diabetes highlights the systemic nature of IMIDs. As ustekinumab has shown efficacy in reducing inflammation and preserving insulin function, it could play a role in managing psoriasis patients who are at high risk of developing metabolic syndrome and type 1 diabetes. The strong negative association between hormone secretion/obesity-related genes and psoriasis severity suggests that molecular phenotyping could usefully be built into future therapeutic studies that target both metabolic (e.g. Glucagon-like peptide−1 agonists) and inflammatory pathways.

The blood RNA-seq data analysis in psoriasis subjects presented here is also the most extensive so far, both in terms of sample number and depth and breadth of variables explored; previous studies have only used data from 6 psoriasis subjects[57] or 23 subjects with psoriatic arthritis[58]. Our ICA

analysis indicated that factor B8 (inflammatory and HLA-aligned B, antigen processing and presentation) and factor B19 (inflammatory and HLA-aligned C, antigen processing and presentation) modules associated with HLA-Cw6 in a positive and negative manner, respectively. While we do not fully understand the relationship between the Cw6 allele and psoriasis susceptibility, a number of mechanisms have been proposed by which the HLA-Cw6 antigen could directly initiate an immune response[59]. The link between HLA-Cw6 and expression patterns in the 6p21 region could represent an additional mechanism by which the allele influences the development of psoriasis. Of the genes enriched in the factor B8 module, class II *HLA-DRB1* has been associated with psoriasis susceptibility and may be druggable[60]. Of the genes associated with the factor B10 module, *HLA-DRA* encodes the invariant α-chain partner to DRB1, playing a central role in antigen presentation, and *HLA-DMB* is crucial for MHC class II peptide loading.

Although Cw6 status has been linked to expression of individual genes, including *IL-1α* and *IL-6*[61], systems-level expression signatures associated with Cw6 status have not previously been identified, and whilst previous studies have found evidence connecting *HLA-Cw6* to disease presentation and severity[59], the Cw6-associated transcriptomic factors we have identified in blood were not correlated with disease severity. Therefore, these signatures likely indicate markers of psoriasis susceptibility, rather than severity.

Not surprisingly, the association of eigengene expression of modules in blood with cutaneous disease activity was less strong compared to modules in lesional skin. Nevertheless, the khaki (innate immune cell; neutrophil degranulation) module and factor B1 (inflammatory and HLA aligned A; phagosome) were positively and significantly associated with disease severity, but only in the adalimumab cohort. Systems-, deconvolution- and gene-level analysis indicated enrichment of neutrophils and genes related to neutrophil degranulation, consistent with previous reports that the blood transcriptional signature of psoriasis is neutrophil driven[40]. The data reported here suggest that neutrophil activation in the blood is linked to disease activity and is modified by adalimumab during therapy. Gene-level analysis in blood was consistent with our findings at a modular level, with significant disease-associated genes only being found in subjects treated with adalimumab. For example, increased disease severity-associated expression of the G-protein coupled receptor, ADGRG3, expressed by tissue-infiltrating granulocytes in blood, but not in lesional or nonlesional skin, is consistent with its known relative tissue specificity towards the cardiovascular system and liver[62]. Together, our findings underscore neutrophils as an important therapeutic target of adalimumab, with expression of key neutrophil degranulation genes in blood reflecting psoriasis disease severity in the skin, and suggest that this signature is linked to TNF.

Genetic studies have identified genetic risk factors in the major histocompatibility region (MHC) for psoriasis that are independent of HLA-Cw6, including variants within the HLA-DQA1 gene[63]. In the present study, we discovered three clusters of subjects based on the expression of the genes *HLA-DQA1*, *HLA-DQB1*, *HLA-DRA*, *HLA-DRB1* and *HLA-DRB5*. These clusters appeared to be determined by the HLA-DQA1*01 and HLA-DRB1*15 genotypes, and were associated with PASI at baseline (Fig. 5). A transcriptome-wide association study found associations between three of these genes (*HLA-DQA1*, *HLA-DQB1* and *HLA-DRB1*) and psoriasis[64]. Furthermore, HLA-DQA1*01:04 has previously been shown to be more common in patients with psoriasis[65]. Interestingly, a specific haplotype comprising HLA-DRB1, -DQA1 and -DQB1 has been linked to protection against the development of type 1 diabetes[66]. Taken together, these results suggest that a haplotype combining HLA-DQA1*01 and HLA-DRB1*15 genotypes could mediate psoriasis susceptibility and severity through the expression of a small number of HLA genes. These genes encode MHC class II surface proteins that are expressed by professional antigen-presenting cells, including Langerhans cells, with key roles in the maintenance of self-antigen tolerance. Their link with psoriasis may help explain the presence of autoreactive T cells in psoriatic tissue[67].

The complex interplay of a multitude of underlying drivers of biologic responses means that the relationships between gene expression and clinical parameters such as disease severity may be best described by non-linear relationships. Given the relatively limited number of patients available (for machine learning) and incomplete understanding of prior relationships, a Gaussian Process Regression (GPR) probabilistic model was explored; GPR has been shown to model numerous complex biological relationships, such as protein-function relationships, with a limited amount of data[23,24,67,68,69]. We used a linear ridge regression (RR) model as a baseline comparison. Although both GPR and RR modelled disease severity with similar accuracy, GPR was more selective than RR, in part due to its ability to flexibly model features more independently as a sum of functions rather than aggregating features and modelling them as part of a single function, as is done in the RR. Overall, our model's performance explains, on average, half of the variation in disease severity. The probabilistic properties of the Bayesian GPR model provide uncertainty estimates, allowing the assessment of unseen patients even in the absence of testing data, which can be further explored in future work.

We identified disease severity-associated gene expression profiles in skin and blood through association with PASI, which captures disease activity across the whole body. Given that the skin transcriptome data were derived from small (5 mm) lesional and nonlesional biopsies predominantly taken from the lower back, our findings support a coordinated disease response that represents integration of signals across cutaneous and circulating compartments.

The majority of modules and factors associated with disease severity in lesional skin were independent of biologic treatment class, in part reflecting clinical response and clearance of cutaneous psoriasis induced by both adalimumab and ustekinumab over 12 weeks. These findings are consistent with those of Brodmerkel et al.[70], who showed convergence of gene expression in response to ustekinumab and etanercept (another TNFi) over 12 weeks.

Of the genes associated with disease severity for both drugs, the majority were assigned to the turquoise (which was closely aligned to factors S1 and S2) and blue modules (which were closely aligned to factors S9 and S22). Systems analysis showed that the turquoise module comprises genes related to pathways and processes with key roles in psoriasis pathogenesis, including cytokine signalling, antimicrobial peptides and cornification (e.g. S100A8/A9, IFN gamma, IRF7, STAT3, CXCL1, DEFB4A, IL-17A/F, IL-36A, PI3, IVL and SPRR1A).In line with the current understanding of the role of innate and acquired immunity in psoriasis pathogenesis[1], bulk RNA-seq cell deconvolution demonstrated that turquoise (cytokine signalling) correlated most closely with myeloid-1 (dendritic cells/macrophages), T cell-2 (Th1, Th17), T cell-3 (cytotoxic T lymphocytes), myloid-1, venule-2 cells and keratinocyte- 1, 4, (spinous) 6 (suprabasal) and 7 (basal) subsets. In line with recent literature, highlighting the potential role of inflammatory fibroblasts in psoriasis pathophysiology[71], the blue module (WNT signalling) included genes such as BTC, an epidermal growth factor receptor (EGFR) ligand which regulates skin homoeostasis and growth. Notably, 100% of blue (WNT signalling) module genes were negatively associated with disease activity. CYP2W1, an "orphan" member of the cytochrome P450 monooxygenase family may play a role in retinoid and phospholipid metabolism[72,73] and has previously been characterised as one of the top down-regulated genes in psoriasis[74]. Metascape analysis placed BTC within the matrisome and deconvolution localised expression to fibroblast-5, fibroblast 2 (universal/reticular)[35,38], keratinocyte-1 (spinous), 5 (supra-spinous) and VSMC cells (Supplementary Fig. 6, keratinocyte-1 (spinous), 5 (supra-spinous) and VSMC cells (Supplementary Fig. A). Network analysis with IPA revealed BTC was associated with the S100 signalling pathway and was upregulated by CD36, along with CCL27, a skin-specific memory T-cell chemokine that is reduced in lesional psoriasis, facilitating IL-17/−22 T-cell overactivity[75] (Fig. 6e, Supplementary Fig. 16). CD36 is a cell-surface scavenger receptor with numerous functions including the regulation of angiogenesis, innate immunity and lipid metabolism[76].

Using GPR and SHAP, we identified a 9-gene signature driving PASI prediction (Fig. 4d). This signature comprised 4 genes in the turquoise module, positively associated with PASI: *CARHSP1*, *KLK13*, *GJB2* and *CRABP2*; interestingly, these have all been associated with psoriasis previously[76–80]; to our knowledge, none have been associated with psoriasis disease severity. The remaining 5 genes in the signature mapped to the blue module and were negatively associated with PASI: *THRA*, *ZNF34*, *RORC*, *CRY2* and *CACNA2D2*. *THRA* encodes for Thyroid Hormone Receptor Alpha and is thought to play a role in healthy skin development and homoeostasis[81]. This is the first report of an association between *THRA*, *ZNF34* and *CACNA2D2* with psoriasis and underscores the potential for novel drug targets amongst genes negatively associated with PASI. Notably, *RORC* and *CRY2* are core clock genes. It has previously been estimated that the circadian clock - representing a group of autoregulatory transcription factors - governs the rhythmic expression of half of all the genes in the human body[82]. Circadian rhythms are disrupted in several inflammatory diseases including psoriasis, in which clock gene expression, including *CRY2* but not *RORC*, has been shown to be disrupted[83]. A core gene signature, comprising a limited number of genes, is more easily assayed than a gene module (which might comprise hundreds, or even thousands, of genes) and therefore represents a potential biomarker that can be tested for clinical utility.

Strikingly, the expression levels of several modules in nonlesional skin were also significantly correlated with PASI. This is interesting in light of work by Gudjonsson et al.[39], which found that nonlesional skin is transcriptionally "pre-psoriatic", and suggests that reduction in disease activity over the course of treatment is accompanied by systems-level changes in expression in nonlesional skin as well as lesional skin and further underscores the systemic nature of psoriasis beyond the visible plaques.

In nonlesional skin, and to an even greater extent in blood, there was a much greater disparity between the drug groups with respect to disease severity-associated modules, factors and genes. These findings suggest that adalimumab, by targeting the pleiotropic cytokine TNF, leads to systemic transcriptome modulation in nonlesional skin and blood (in addition to lesional skin), whereas the transcriptome changes induced by ustekinumab through inhibition of IL-12/23 are largely restricted to lesional skin. Moreover, this disparity with greater modulation of both innate and acquired immune pathways in blood by adalimumab compared to ustekinumab is consistent with a relative increase in secondary infections, including TB, with adalimumab compared to ustekinumab[44].

This experimental medicine study utilised the first classes of biologics introduced for psoriasis. Although there are now newer classes of biologics available that are more effective for managing psoriasis, adalimumab and ustekinumab continue to represent the most widely prescribed biologics in clinical practice in many countries, including the UK[43]. Although our study of the psoriasis blood transcriptome is the largest reported to date, and reproduces a neutrophil signature[40], further independent validation analysis is required with respect to the disease severity endotype identified exclusively in the blood of patients treated with adalimumab. In contrast to our combined analysis of lesional and nonlesional psoriasis, our separate analysis of skin and blood precluded direct comparisons between these tissue compartments.

The systematic nature of the dataset across skin and blood and its integration with key clinical features and outcomes to therapy provides a comprehensive and global perspective of psoriasis that are not apparent from previous studies. Prior pharmacogenomic evaluations of patient cohorts have centred on the use of genetic or genomic techniques, predominantly using skin biopsies, although several studies have used skin and blood[57,84,85]. Importantly, no prospective biomarkers of disease severity or response have yet been validated in adequately powered cohorts[86]. Our results are fully reproducible via companion R markdown documents and, as in our pilot investigation, we encourage readers to replicate our analysis in independent cohorts, and we continue to advocate the open access of data and code for all studies to improve reproducibility and drive excellent science[5]. Our workflow can be applied to evaluate datasets of disease severity

and clinical response to novel therapies outside of dermatological indication. Further work is required to derive scalable biomarkers for testing in the clinical setting. Integrated analysis of samples across drug cohorts, time points, and lesional and non-lesional skin when carrying out WGCNA and ICA greatly reduced the dimensionality of the dataset. However, this approach precluded the identification of modules and factors which might be specific to lesional or non-lesional skin, or to certain time points; future work might involve the identification of modules and factors separately in these different subsets of samples. Methods such as module preservation analysis for WGCNA, which has been used in this study to compare skin and blood (Supplementary Fig. 4), could then be used to assess the degree to which module co-expression changes across tissues and time points. Crucially, differences in co-expression may be separate to differences in absolute expression levels, which are the focus of our endotype analyses in this paper and more suited to the integrated dimensionality reduction approach taken here.

The authors believe it will be standard practice that molecular phenotyping will drive personalised therapy across therapeutic targets for immune-mediated inflammatory diseases in the future. Potential limitations of the study include a lack of randomisation of subjects to treatment arms (although the study cohorts were well-matched) and the absence of treatment adherence assessment (although the response rates seen suggest that participants were adherent to therapy). The insights provided here of the complex interface between these diseases and disease severity endotypes provide a framework for delineating molecular and clinical response across tissues and predicting individual subject response in future studies.

## Data availability

Sharing of (personally identifiable) raw sequencing data and detailed clinical data is limited by institutional and UK ethical, consent, privacy and governance frameworks. The raw and adjusted gene count data from our RNA-seq analysis are available at Array Express under accession number E-MTAB-14509 with associated clinical data, specifically PASI and BMI. This approach allows us to share valuable processed data for replication, validation and further analysis while respecting participant privacy and adhering to governance and ethical guidelines. The supplementary data files, in conjunction with the data on Array Express and the code on GitHub, should facilitate full reproduction of figures. The supplementary data is provided in a number of Excel documents: "Supplementary Data 1. WGCNA module data.xlsx" and "Supplementary Data 2. ICA factor data.xlsx", which are required for Figs. 2–7; "Supplementary Data 3. Module and factor-trait correlations.xlsx", which is required for Fig. 2; "Supplementary Data 4. Metascape results for module and factor genes.xlsx" and "Supplementary Data 5. Module and factor metadata.xlsx", which were used to derive the module and factor descriptors displayed in Fig. 2a; "Supplementary Data 6. PASI differential expression results.xlsx", which is required for Figs. 3, 6 and 7; "Supplementary Data 7. Expression data for example PASI DEGs.xlsx", which is required for Figs. 3 and 6; "Supplementary Data 8. Metascape results for PASI DEGs.xlsx", which is required for Fig. 7d; "Supplementary Data 9. IPA results for PASI DEGs.xlsx", which is required for Figs. 6e and 7e; "Supplementary Data 10. BMI differential expression results.xlsx", which is required for Fig. 3; "Supplementary Data 11. HLA endotype data.xlsx", which is required for Fig. 5; "Supplementary Data 12. GPR - PASI prediction of Gaussian process model using skin modules.xlsx" and "Supplementary Data 13. GPR—PASI prediction of Gaussian process model using skin factors.xlsx", which are required for Fig. 4a; "Supplementary Data 14. GPR—Feature importance for skin modules.xlsx" and "Supplementary Data 15. GPR—Feature importance for skin factors.xlsx", which are required for Fig. 4b; "Supplementary Data 16. GPR—PASI prediction importance for turquoise and blue genes.xlsx", which is required for Fig. 4c; and "Supplementary Data 17. GPR - Gene signature from turquoise and blue.xlsx", which is required for Fig. 4d. The predicted cell type fractions from single cell deconvolution analysis are also available on GitHub (https://github.com/C4TB/PSORT/tree/master/Cell_Type_Correlations/paper_

data). Additionally, the PSORT skin and blood RNA-seq dataset may be visualised and further explored through an R Shiny web interface[42] (https://shiny-whri-c4tb.hpc.qmul.ac.uk/psort).

## Code availability

Data analysis scripts can be found on our GitHub repository (https://github.com/C4TB/PSORT), along with extended supplemental markdown documents. These are also deposited[87] on Zenodo (https://doi.org/10.5281/zenodo.15847636).

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

## Acknowledgements

Non-author contributions. We are grateful to all subjects for their participation. We thank the Independent Advisory Board who provided valuable advice and an independent international stakeholder perspective to the PSORT Consortium (Iain McInnes [Chair], John Armstrong, Anne Bowcock, James Krueger, Christy Langan and Peter van de Kerkhof). We are grateful to the Psoriasis Association for their Patient and Public Involvement and Engagement. We acknowledge the enthusiastic collaboration and support of dermatologists, specialist and research nurses in the UK who recruited to this study including Alberto Barea (Kingston Hospital NHS Foundation Trust), Dr Anna Chapman (Lewisham & Greenwich Trust), Dr Rob Ellis (South Tees Hospitals NHS Foundation Trust), Dr Abigail Fogo (Kingston Hospital NHS Foundation Trust), Dr Bronwyn Hughes (Portsmouth Hospitals University NHS Trust), Dr Evmorfia Ladoyanni (Dudley Group NHS Foundation Trust), Dr Philip Laws (Leeds Teaching Hospitals NHS Trust), Dr Richard Parslew (Liverpool University Hospitals NHS Foundation Trust), Dr Gayathri Perera (Chelsea and Westminster Hospital NHS Foundation Trust), Dr Beth Poyner (Newcastle upon Tyne Hospitals NHS Foundation Trust), Sara Wilkinson (Newcastle upon Tyne Hospitals NHS Foundation Trust). We thank Hira Ali, Rosa Andres-Ejarque, Zaynep Catak, Tejus Dasandi, Nadya Dinev, Michael Duckworth, Katarzyna Grys, Freya Meynell, Alice Russel and Isabella Tosi (London), and Dhanisha Lukka and Panagiotis Maniatis (Newcastle) for sample and data management. We thank Federica Villanova (London) for her contribution to obtain ethical approval. Frank Nestle contributed to the initial stages of this work and the authors wish to acknowledge his contribution. This study was funded by The Medical Research Council (MRC)(MICA MRC Precision Medicine Consortium; MR/L011808/1); The Psoriasis Association; The British Association of Dermatologists (BAD); The Rosetrees Trust; UCB; The NIHR Newcastle Biomedical Research Centre (BRC); and a MRC/BAD/British Skin Foundation clinical research training fellowship to HJG. Partners of the PSORT consortium are AbbVie, the British Association of Dermatologists, Becton Dickinson and Company, Celgene Limited, GlaxoSmithKline, Guy's and St Thomas' NHS Foundation Trust, Eli Lilly, Janssen Research & Development, King's College London, LEO Pharma, MedImmune, Novartis Pharmaceuticals UK, Pfizer Italy, the Psoriasis Association, Qiagen Manchester, Queen Mary University of London, the Royal College of Physicians, Sanquin Blood Supply Foundation, the University of Liverpool, the University of Manchester, and Newcastle University. We particularly acknowledge generous in-kind support from the PSORT industrial partners GSK and Abbvie who supported the RNA sequencing. All decisions concerning analysis, interpretation, and publication are made independently of any of the 12 industrial contributions. NJR's research/laboratory is funded in part by the NIHR Newcastle HealthTech Research Centre in Diagnostic and Technology Evaluation and the NIHR Newcastle Patient Safety Research Collaboration. N.J.R. is a NIHR Senior Investigator. P.Z. received funding from Sapienza University (#RP123188F6A67836) CEMG and RBW are funded in part by the Manchester NIHR BRC (NIHR203308). M.R.B. and D.W. were funded by the NIHR as part of the portfolio of translational research of the NIHR BRC at Barts and The London School of Medicine and Dentistry. C.S. receives research funding from European consortia with multiple industry partners (see BIOMAP-imi.eu and HIPPOCRATES-imi.eu, MRC-industry PhD studentships from AstraZeneca, Boehringer Ingelheim). This project was enabled through access to the MRC eMedLab Medical Bioinformatics infrastructure supported by the Medical Research Council [grant number MR/L016311/1].

## Author contributions

A.R., H.J.G., G.R.S. and D.S.W. contributed equally as joint first authors. J.C., S.J.C., J.G., A.C.F., R.H. and W.A.I. contributed equally as joint second authors. J.N.B., C.E.M.G., P.D.M., C.H.S. and R.B.W. contributed equally as joint penultimate authors. M.R.B. and N.J.R. jointly supervised the study. Conceptualisation: M.R.B., J.N.B., C.E.M.G., N.J.R., C.H.S., D.S., R.W. Data Curation: A.R., D.S.W., N.D. Formal Analysis: H.J.G., A.R., D.S.W., G.R.S., J.C., J.G., S.J.C., S.N., W.A.I. Funding acquisition: M.R.B., J.N.B., C.E.M.G., N.J.R., C.H.S., D.S., R.W., P.Z. Investigation: A.C.F., S.K.S., T.E., C.T., E.T., D.K.R., K.M.S., J.N.B., C.E.M, C.H.S., R.W., N.J.R. Methodology: D.S., S.K.S., C.T., E.T., D.K.R., K.M.S., W.A.I., N.D., S.A. Project administration: M.R.B., J.N.B., P.D.M., C.E.M.G., N.J.R., C.H.S., R.W. Resources: M.R.B., J.N.B., C.E.M.G., N.J.R., C.H.S., R.W. Software: R.H. Supervision: M.R.B., N.J.R Validation: H.J.G., A.R., G.R.S., J.C., J.G., W.A.I. Visualisation: H.J.G., A.R., G.R.S., J.C., J.G., W.A.I. Writing—original draft: H.J.G., A.R., J.G., W.A.I., M.R.B., N.J.R. Writing—review & editing: all authors.

## Competing interests

N.J.R. has received, through Newcastle University, grants from Novartis and UCB outside the submitted work; travel support and/or consultancy income from AbbVie, Boehringer Ingelheim, Galderma, and UCB. M.R.B. has received research funding from Janssen and Benevolent AI, and consultancy with United Health group, Eli Lilly and Sanofi. All other authors declare no competing interests.

## Additional information

[1]Institute of Translational and Clinical Medicine, Faculty of Medical Sciences, Newcastle University, Newcastle upon Tyne, UK. [2]Bioinformatics Support Unit, Faculty of Medical Sciences, Newcastle University, Newcastle upon Tyne, UK. [3]Centre for Translational Bioinformatics, William Harvey Research Institute, Queen Mary University of London, Charterhouse Square, London, UK. [4]Department of Informatics, King's College London, London, UK. [5]Biosciences Institute, Faculty of Medical Sciences, Newcastle University, Newcastle upon Tyne, UK. [6]The Manchester Centre for Dermatology Research, The University of Manchester, Manchester, UK. [7]School of Computing, Newcastle University, Newcastle upon Tyne, UK. [8]Dipartimento di Informatica, Sapienza University, Rome, Italy. [9]Department of Medical and Molecular Genetics, Kings College London, London, UK. [10]Leeds Institute of Clinical Trials Research, The University of Leeds, Leeds, UK. [11]Computational Biology, GlaxoSmithKline, Collegeville, PA, USA. [12]Takeda Development Center Americas, Inc. (TDCA), Lexington, MA, USA. [13]AbbVie Genome Research Center, Immunology Bioinformatics, 200 Sidney Street, Cambridge, MA 02139, USA. [14]St. Johns Institute of Dermatology, Kings College London, London, UK. [15]NIHR Newcastle Biomedical Research Centre & Department of Dermatology, Royal Victoria Infirmary, Newcastle Hospitals NHS Foundation Trust, Newcastle upon Tyne, UK. [16]These authors contributed equally: Ashley Rider, Henry J. Grantham, Graham R. Smith, David S. Watson. [17]These authors jointly supervised this work: Michael R. Barnes, Nick J. Reynolds. ✉e-mail: m.r.barnes@qmul.ac.uk; nick.reynolds@newcastle.ac.uk

## On behalf of the PSORT consortium

Jonathan N. Barker[14], Christopher E. M. Griffiths[6], Paola Di Meglio[14], Catherine H. Smith ✪[14], Richard B. Warren[6], Michael R. Barnes ✪[3,17]✉ & Nick J. Reynolds ✪[1,15,17]✉

A full list of members and their affiliations appears in the Supplementary Information.

