## [Transparent Peer Review file · Communications Medicine]

Transcriptomic profiling and machine learning uncover gene signatures of psoriasis endotypes and disease severity

Corresponding Author: Professor Nick Reynolds

Version 0:

Reviewer comments:

Reviewer #2

(Remarks to the Author)

With this study Reynolds and co-workers provide a deeper understanding of the pathogenetic mechanisms of psoriasis and could help develop more personalized therapies for patients.

They used multiomics technology on blood, non lesional and lesional biopsies of patients before and after treatment with the anti-TNF drug adalimumab or the anti-IL12/23 drug Ustekinumab and:

- 1) Genetic signatures associated with disease severity have been identified in lesional and nonlesional skin, as well as in blood. Two main modules in lesional skin were found, one positively and one negatively associated with disease severity, regardless of treatment with adalimumab or ustekinumab.
- 2) A 14-gene genetic signature has been identified in nonlesional skin, associated with body mass index (BMI) and negatively correlated with disease severity in lesional skin.
- 3) Blood disease severity signatures were specific to treatment with adalimumab, suggesting a greater systemic impact of this drug compared to ustekinumab.
- 4) Integration of genotype data identified the HLA-DQA101 and HLA-DRB115 genotypes as associated with disease severity at baseline.
- 5) The results outline the combined effects of genetic and environmental factors on the psoriasis transcriptome, linked to disease severity.

The manuscript is well written, and I could not identify any major flaw.

However, given the amount of information delivered by this study, the manuscript is a bit heavy and difficult to read. I suggest the inclusion of graphical representation of key results among the main figure, putting some of the graphs in the supplementary section.

Reviewer #3

(Remarks to the Author)

Rider et al. perform a differential gene expression analysis of transcriptomes from skin biopsies and blood samples. The study presents convincing, though complex results that define certain molecular subtypes of psoriasis. Strengths of the study are the size of the study groups, the combination of a discovery and replication data set, the in-depth analysis of endotypes, the multidimensional analysis of the complex data and the time line of several biopsies performed during the initiation of the systemic treatment. However, a few aspects need some exploration and clarification:

- The description of the cohorts indicates that in the discovery group each individual had $n=9$ samples, resulting in an expected sum of 783 RNASeqs, while the reported ones are 718. The numbers in the replication cohort are $n=5$ in 57 individuals resulting in a sum of 285 RNASeqs. The percentages of missing RNASeqs are different between the groups: 92% vs. 97%. Did the authors test whether these missing data confound the result? Is there an even distribution of missing samples at different time points? If not, has anything been done to take that into account?
- The study took a timely manner of gene expression following the beginning of a systemic treatment into account. This is an important part of the study; however, the study does not comment on the course of changes of gene expression/ gene modules/ factor metagenes. Did the study not reveal significant findings here?
- Regarding clinical data, it is not obvious which ones beside BMI and PASI were assessed in the study.
- The authors show that some the expression of some modules/ gene sets overlap between blood and skin. Different

mechanistic aspects of psoriasis in skin (barrier function) and blood (adaptive immune system) have been shown to be relevant in the pathogenesis psoriasis and contribute to its manifestation. However, which remarkable gene expression patterns/ modules are rather specific for skin or blood that represent new findings?

- Associations were observed for several (disease) endotypes with e.g. obesity-related modules and BMI, thereby replicating findings of previous studies. A consequence of this finding would indicate that not only potential therapeutic strategies that target both metabolic and inflammatory pathways (lines 489-490) will be beneficial, but also motivate affected individuals to lose weight. Moreover, the authors state that the transcriptomic profiling might be helpful for the prognosis of other affected individuals in daily practice. However, a direct consequence for affected individuals would be to lose weight. If the two standard therapeutic strategies, assessed deeply in this study, do not affect the disease endotype or different psoriatic subtypes, it remains rather blurred how and why a molecular phenotyping will drive personalized therapy? This aspect would need further explanation.

Reviewer #4

(Remarks to the Author)

This manuscript presents a comprehensive and well-executed study that integrates transcriptomic profiling with machine learning to uncover gene expression patterns associated with psoriasis endotypes and disease severity. The key strength of the work lies in its dual use of unsupervised (WGCNA and ICA) and supervised (Gaussian Process Regression) machine learning methods, alongside SHAP interpretability analyses. These approaches allow for both hypothesis-free clustering and interpretable prediction modeling, which adds depth and translational relevance to the findings.

The major claims—that transcriptomic signatures can stratify patients into biologically meaningful clusters and that disease severity can be predicted with reasonable accuracy—are well-supported by the data and analysis. These findings are novel and will be of clear interest to the dermatology and systems biology communities. The manuscript is likely to influence ongoing research in patient stratification and AI applications in inflammatory diseases.

A few suggestions may improve clarity and impact:

Justification of Predictive Model Selection: While Gaussian Process Regression is an elegant and probabilistic method, it would be helpful for the authors to briefly explain why GPR was chosen over other models such as random forests, gradient boosting, or neural networks, particularly in the context of transcriptomic data.

Reproducibility and Code Availability: The authors provide a well-structured methodological description. However, full reproducibility would be enhanced by depositing code and processed data into a public repository (e.g., GitHub, Zenodo), particularly for the machine learning pipeline and SHAP interpretability scripts.

Biological Interpretation: The authors have identified module genes associated with distinct endotypes. A more detailed discussion connecting these modules to known psoriasis biology or prior literature could help contextualize the findings for readers.

Statistical Robustness: The use of SHAP values to interpret model predictions is commendable and appropriate. Please confirm whether any cross-validation (e.g., k-fold) or permutation testing was performed to validate model generalizability and avoid overfitting.

Visualizations: Some figures (e.g., ICA plots) could benefit from clearer labeling or explanations in the legends, especially for readers less familiar with the methods.

In summary, this is a strong manuscript that combines methodological rigor with clinical relevance. With a few minor clarifications and enhancements, it should make a valuable contribution to the field of precision dermatology and explainable machine learning in biomedicine.

Version 1:

Reviewer comments:

Reviewer #1

(Remarks to the Author)

The authors have satisfactorily addressed my concerns.

Reviewer #3

(Remarks to the Author)

The authors convincingly demonstrate that transcriptomic signatures can stratify patients into biologically meaningful clusters and predict disease severity with reasonable accuracy. The use of both unsupervised methods (WGCNA, ICA) and supervised learning (Gaussian Process Regression with SHAP interpretability) is novel and powerful. These findings are

original, align with current challenges in precision dermatology, and will be of interest not only to dermatology researchers but also to the broader systems biology and translational medicine communities.

Overall, this is a high-quality and comprehensive manuscript. The authors have satisfactorily addressed all previous comments, and I support publication.

Signed:
Professor Yeon-Hee Lee
Kyung Hee University, School of Dentistry

Comments	Responses
Reviewer comments: Reviewer: 1 Comments to the Author With this study Reynolds and co-workers provide a deeper understanding of the pathogenetic mechanisms of psoriasis and could help develop more personalized therapies for patients. They used multiomics technology on blood, non lesional and lesional biopsies of patients before and after treatment with the anti-TNF drug adalimumab or the anti-IL12/23 drug Ustekinumab and: 1) Genetic signatures associated with disease severity have been identified in lesional and nonlesional skin, as well as in blood. Two main modules in lesional skin were found, one positively and one negatively associated with disease severity, regardless of treatment with adalimumab or ustekinumab. 2) A 14-gene genetic signature has been identified in nonlesional skin, associated with body mass index (BMI) and negatively correlated with disease severity in lesional skin. 3) Blood disease severity signatures were specific to treatment with adalimumab, suggesting a greater systemic impact of this drug compared to ustekinumab. 4) Integration of genotype data identified the HLA-DQA101 and HLA-DRB115 genotypes as associated with disease severity at baseline. 5) The results outline the combined effects of genetic and environmental factors on the psoriasis transcriptome, linked to disease severity.	
The manuscript is well written, and I could not identify any major flaw. However, given the amount of information delivered by this study, the manuscript is a bit heavy and difficult to read. I suggest the inclusion of graphical representation of key results among the main figure, putting some of the graphs in the supplementary section.	Thank you for this comment. We have amended Fig 1 to provide a clearer overview of the workflow and key findings in the paper. To streamline the paper, we have moved some data into supplementary:  • The module-factor cross correlation heatmaps have been removed from Fig. 2 and moved to Supplementary Fig 3.

	 • We have reduced the number of exemplar module/factor-trait correlation plots shown in Fig.2. Further exemplar plots are shown in Supplementary Fig. 9. • The heatmap showing BMI and PASI associations of genes assigned to WGCNA modules has been moved from Fig. 3 to Supplementary Fig. 5. • The non-lesional skin data from Fig. 7 has been moved to Supplementary Fig. 18. We have also altered figure 2 so that the exemplar module/factor-trait correlation plots are organised into meaningful groups. We believe this has strengthened the manuscript as the main figures are more clearly focussed on the main messages of the paper.
Reviewer: 2 Rider et al. perform a differential gene expression analysis of transcriptomes from skin biopsies and blood samples. The study presents convincing, though complex results that define certain molecular subtypes of psoriasis. Strengths of the study are the size of the study groups, the combination of a discovery and replication data set, the in-depth analysis of endotypes, the multidimensional analysis of the complex data and the time line of several biopsies performed during the initiation of the systemic treatment. However, a few aspects need some exploration and clarification:	
- The description of the cohorts indicates that in the discovery group each individual had n=9 samples, resulting in an expected sum of 783 RNASeqs, while the reported ones are 718. The numbers in the replication cohort are n=5 in 57 individuals resulting in a sum of 285 RNASeqs. The percentages of missing RNASeqs are different between the groups: 92% vs. 97%. Did the authors test whether these missing data confound the result? Is there an even distribution of missing samples	The main portion of this missingness arises from a small number of patients for whom only blood (n=7) or skin (n=6) samples were analysed by RNAseq. We provide a summary of the missed samples in Supplementary Fig. 21. We also provide a demographic table for the full patient cohort (i.e. all patients used for analysis) and, for comparison, a demographic table with these 13 patients removed. The demographics are similar between the two tables, suggesting that

at different time points? If not, has anything been done to take that into account?	inclusion of these 13 patients in the analysis did not introduce any systematic bias.
- The study took a timely manner of gene expression following the beginning of a systemic treatment into account. This is an important part of the study; however, the study does not comment on the course of changes of gene expression/ gene modules/ factor metagenes. Did the study not reveal significant findings here?	Thank you for commenting on this. Because of the complexity of the current analysis and findings, we are planning to report on the sequential changes in gene expression with adalimumab and ustekinumab, using the framework reported here. For clarity, we have added a sentence to the discussion (lines 763-764).
- Regarding clinical data, it is not obvious which ones beside BMI and PASI were assessed in the study.	We computed correlations between module eigengenes/ICA factor variables and several clinical variables:  • Age of onset • Onset type (type I or II) • Anti-TNF naive status • Psoriatic arthritis • Sex • Age • HLA-Cw6 • BMI • PASI Please see the module-trait correlations section of the supplementary text. Note that only variables that exhibited a significant and replicable correlation are displayed in figure 2A. We have clarified this in the “Module-trait correlations” section of the supplementary text (lines 203-204).
- The authors show that some the expression of some modules/ gene sets overlap between blood and skin. Different mechanistic aspects of psoriasis in skin (barrier function) and blood (adaptive immune system) have been shown to be relevant in the pathogenesis psoriasis and contribute to its manifestation. However, which remarkable gene expression patterns/ modules are rather specific for skin or blood that represent new findings?	Thank you for the comment. We carried out module preservation analysis using functionality from the WGCNA R package in order to assess the extent to which skin modules were preserved in blood and vice versa. This identified a subset of modules, containing genes involved in oxidative phosphorylation and translation, that exhibited strong evidence for preservation between the two tissue compartments. Text for this analysis has been added to the “Identification of gene expression signatures in skin and blood” section of the results (lines 268-274), along with Supplementary Fig. 4. The turquoise cytokine signalling module comprises classic psoriasis immune and

	innate signalling pathways, including interferons, IL-17, IL-23 and TNF, as well genes involved in skin barrier homeostasis and is positively associated with disease severity in lesional skin (Fig 2, Fig 6). Specific sets of barrier genes show distinct co-regulation and cluster into darkgreen (positively associated with disease activity) and skyblue modules that negatively associate with disease activity. The separation of cornified gene modules into those positively and negatively associated with PASI is new and has been further emphasised in the manuscript (line 328, line 332, lines 462-468). Moreover, modules and genes that may negatively regulate disease activity (e.g. yellow module and factor 9; ECM and Insulin and hormone secretion signalling and blue module (WNT) (e.g. BTC and RORC genes) has been relatively under-explored and may represent novel therapeutic targets. This has been further emphasised in the discussion (lines 515-521). To our knowledge, a systematic investigation of gene modules with disease severity has not previously been reported and in particular the characterisation of modules that negatively correlate with PASI is novel. The finding that eigengene expression of three WGCNA modules: turquoise (e.g. S100A7A), darkgreen (cornification 1) and paleturquoise and factor S1 (Antimicrobial protein; e.g. SPRR2G) in non-lesional-skin significantly and reproducibly positively correlated with whole body disease severity in the adalimumab group (Fig. 2A) is a new finding.
- Associations were observed for several (disease) endotypes with e.g. obesity-related modules and BMI, thereby replicating findings of previous studies. A consequence of this finding would indicate that not only potential therapeutic strategies that target both metabolic and inflammatory pathways (lines 489-490) will be beneficial, but also motivate affected individuals to lose weight. Moreover, the authors state that the transcriptomic profiling might be helpful for the prognosis of other affected individuals in daily practice. However, a direct consequence for affected individuals would be to lose weight. If the two	Thank you for this comment. In this study, we defined gene modules (lightyellow [insulin and hormone secretion] in WGCNA and factor S9 in ICA [obesity-associated) and 14-gene signature in non-lesional skin that negatively correlated with BMI and with disease activity (PASI) at baseline. At lines 586-589, we state “The strong negative association between hormone secretion/obesity-related genes and psoriasis severity suggests that molecular phenotyping could usefully be built into future therapeutic studies that target both

standard therapeutic strategies, assessed deeply in this study, do not affect the disease endotype or different psoriatic subtypes, it remains rather blurred how and why a molecular phenotyping will drive personalized therapy? This aspect would need further explanation.	metabolic (e.g. Glucagon-like peptide (GLP)-1 agonists) and inflammatory pathways.” We agree with the reviewer that further studies are required to more clearly define the potential utility of molecular phenotyping. Additionally, we suggest that such molecular phenotyping could usefully be built into future studies combining biologic or small molecule therapy with weight reduction programs and or GLP agonists.
Reviewer: 3 This manuscript presents a comprehensive and well-executed study that integrates transcriptomic profiling with machine learning to uncover gene expression patterns associated with psoriasis endotypes and disease severity. The key strength of the work lies in its dual use of unsupervised (WGCNA and ICA) and supervised (Gaussian Process Regression) machine learning methods, alongside SHAP interpretability analyses. These approaches allow for both hypothesis-free clustering and interpretable prediction modelling, which adds depth and translational relevance to the findings. The major claims—that transcriptomic signatures can stratify patients into biologically meaningful clusters and that disease severity can be predicted with reasonable accuracy—are well-supported by the data and analysis. These findings are novel and will be of clear interest to the dermatology and systems biology communities. The manuscript is likely to influence ongoing research in patient stratification and AI applications in inflammatory diseases. A few suggestions may improve clarity and impact:	
Justification of Predictive Model Selection: While Gaussian Process Regression is an elegant and probabilistic method, it would be helpful for the authors to briefly explain why GPR was chosen over other models such as random forests, gradient boosting, or neural	Thank you for raising this point. We selected additive Gaussian Process Regression (GPR) over random forests, gradient boosting, and neural networks for the following reasons:

networks, particularly in the context of transcriptomic data.

1. **Small Dataset, High Dimensionality:** With ~250–300 independent observations (139 patients, ≤ 3 time points) and >30,000 transcripts, tree-based models (random forests, XGBoost) and neural networks risk overfitting or underfitting due to numerous hyperparameters and insufficient samples. GPR performs well with limited data (Table 1).
2. **Non-linear Modeling with Uncertainty:** GPR's radial-basis and Matérn kernels capture complex transcriptomic interactions, and its Bayesian framework provides sample-specific prediction intervals, crucial for precision medicine, unlike black-box models.
3. **Interpretability:** GPR's additive structure allows clear feature contributions (e.g., eigengenes) via SHAP analysis (Fig. 4b, d), unlike entangled effects in ensembles or neural networks.
4. **Competitive Performance:** GPR achieved $MAE = 0.45 \pm 0.07$ and $R^2 = 0.53 \pm 0.09$, outperforming ridge regression and matching optimized XGBoost with fewer hyperparameters (Table 1).
5. **Proven Biological Utility:** GPR excels in small-sample, non-linear biological problems like protein fitness mapping (46).

We have commented on this in the discussion section and added the following concise rationale to the Results section (Page 23, lines 360-368), emphasising GPR's balance of accuracy, uncertainty quantification, and interpretability for our transcriptomic dataset.

“To predict disease severity from modules and factors, we employed additive Gaussian-process regression (GPR; Supplementary material and methods). GPR was well-suited to our “small-n, large-p” design, which comprised 718 samples from 146 subjects and approximately 20,000 transcripts within 60 modules and 45 factors

	across skin and blood, as this method combines (i) non-linear flexibility (through radial-basis and Matérn kernels) with (ii) a Bayesian framework that returns per-sample credible intervals, an essential read-out for clinical risk-stratification. Moreover, its additive kernel decomposition facilitated transparent attribution of individual module and factor contributions via SHAP values (Fig. 4b, d), which cannot be obtained as cleanly from tree-based ensembles or neural networks.”
Reproducibility and Code Availability: The authors provide a well-structured methodological description. However, full reproducibility would be enhanced by depositing code and processed data into a public repository (e.g., GitHub, Zenodo), particularly for the machine learning pipeline and SHAP interpretability scripts.	The code is available in a public GitHub repository: https://github.com/C4TB/PSORT and the expression data and associated clinical data are on Array Express under accession MTAB-14509. A R shiny portal is also available online to allow further exploration of results: https://shiny-whri-c4tb.hpc.qmul.ac.uk/psort (username: psort, password: tower squid ramp). Please see the data and materials availability section in the main text. The Array Express dataset is currently set to private and so is not publicly available but will be made open access once the paper is published. Reviewers can view the dataset here https://www.ebi.ac.uk/biostudies/ArrayExpress/studies/E-MTAB-14509?key=38001e0b-15bc-4597-b4e4-09c9310334c8 The password protection of the Shiny app will also be removed upon publication.
Biological Interpretation: The authors have identified module genes associated with distinct endotypes. A more detailed discussion connecting these modules to known psoriasis biology or prior literature could help contextualize the findings for readers.	Thank you for this comment. We have expanded the discussion (lines 669-694, 611-627) to include a more detailed explanation of the gene modules and their links to psoriasis pathophysiology: “Of the genes associated with disease severity for both drugs, the majority were assigned to the turquoise (cytokines) (which was closely aligned to factors S1 and S2) and blue modules (WNT) (which was closely

aligned to factors S9 and S22). Systems analysis showed that the turquoise module comprises genes related to cytokine signalling, antimicrobial peptides and cornification, that are known to play key roles in psoriasis pathogenesis, including S100A8/A9, IFN-gamma, IRF7, STAT3, CXCL1, DEFB4A, IL-17A/F, IL-36A, PI3 and cornification genes (e.g. IVL and SPRR1A) In line with the current understanding of the role of innate and acquired immunity in psoriasis pathogenesis (1), bulk RNAseq cell deconvolution demonstrated that turquoise (cytokine signalling) correlated most closely with myeloid-1 (dendritic cells/macrophages), T cell-2 (Th1, Th17), T cell-3 (cytotoxic T lymphocytes), myeloid-1, venule-2 cells and keratinocyte- 1, 4, (spinous) 6 (suprabasal) and 7 (basal) subsets. In line with recent literature, highlighting potential role of inflammatory fibroblasts in psoriasis pathophysiology (51), the blue module (WNT signalling) included genes such as BTC, an epidermal growth factor receptor (EGFR) ligand which regulates skin homeostasis and growth. Notably 100% blue (WNT signalling) module genes were negatively associated with disease activity. CYP2W1, an “orphan” member of the cytochrome P450 monooxygenase family may play a role in retinoid and phospholipid metabolism(51, 52) and has previously been characterised as one of top down-regulated genes in psoriasis (54). Metascape analysis placed BTC within the matrisome and deconvolution localised expression to fibroblast-5, fibroblast 2 (universal/reticular) (12,15), keratinocyte-1 (spinous), 5 (supra-spinous) and VSMC cells (Supplementary Fig. 6, keratinocyte-1 (spinous), 5 (supra-spinous) and VSMC cells (Supplementary Fig. A). Network analysis with IPA revealed BTC was associated with the S100 signalling pathway and was upregulated by CD36, along with CCL27 (a skin-specific memory T-cell chemokine that is reduced in lesional psoriasis, facilitating IL-17/-22 T-cell overactivity(55)) (Fig. 6e, Supplementary Fig. 16). CD36 is a cell-surface scavenger receptor with numerous functions including

	the regulation of angiogenesis, innate immunity and lipid metabolism (56).” “Not surprisingly, the association of eigengene expression of modules in blood with cutaneous disease activity was less strong compared to modules in lesional skin. Nevertheless, the khaki (innate immune cell; neutrophil degranulation) module and factor B1 (inflammatory and HLA aligned A; phagosome) were positively and significantly associated with disease severity but only in the adalimumab cohort. Systems, deconvolution and gene level analysis indicated enrichment of neutrophils and genes related to neutrophil degranulation, consistent with previous reports that the blood transcriptional signature of psoriasis is neutrophil driven (17). The data reported here suggests that neutrophil activation in the blood is linked to disease activity and is modified by adalimumab during therapy. Gene level analysis in blood was consistent with our findings at a modular level with significant disease associated genes only being found in subjects treated with adalimumab. For example, increased disease severity-associated expression of the G-protein coupled receptor, ADGRG3, expressed by tissue-infiltrating granulocytes in blood but not in lesional or nonlesional skin is consistent with its known relative tissue specificity towards the cardiovascular system and liver (40). Together, our findings underscore neutrophils as an important therapeutic target of adalimumab with expression of key neutrophil degranulation genes in blood reflecting psoriasis disease severity in the skin and suggest that this signature is linked to TNF.”
Statistical Robustness: The use of SHAP values to interpret model predictions is commendable and appropriate. Please confirm whether any cross-validation (e.g., k-fold) or permutation testing was performed to validate model generalizability and avoid overfitting.	We certainly agree that evaluating the performance of models on held out data is essential to assess the model's generalisability. In this study, we employed repeated 10-fold cross-validation (CV) at the subject level. Specifically, the 139 subjects from the combined discovery and replication

	cohorts (PSORT-D and PSORT-R) were randomly allocated to training (80%), validation (10%), and testing (10%) datasets across 20 shuffled 10-fold CV iterations. This approach ensured that all samples from a single subject were kept within the same fold to avoid data leakage, enhancing the robustness of the model's performance estimates. The model's performance was assessed using mean absolute error (MAE = 0.45 ± 0.07) and coefficient of determination ($R^2 = 0.53 \pm 0.09$) on the test sets, as reported in Table 1. These metrics, derived from multiple CV iterations, confirm the model's generalisability across unseen data. This process was alluded to in the methods of our original draft but we did not provide enough detail to adequately describe the procedure. We have now improved this description in the methods (Section Machine Learning, lines 189-205) and supplemental methods.
Visualizations: Some figures (e.g., ICA plots) could benefit from clearer labeling or explanations in the legends, especially for readers less familiar with the methods.	Thank you for this comment. To address this, we have amended a number of figures (e.g. Figure 1), re-written and added further explanation to the main figure legends and defined abbreviations where appropriate. We believe that these changes will make the findings more accessible to readers, particularly those unfamiliar with the interpretable dimensionality reduction techniques we employ in this study.
In summary, this is a strong manuscript that combines methodological rigor with clinical relevance. With a few minor clarifications and enhancements, it should make a valuable contribution to the field of precision dermatology and explainable machine learning in biomedicine.	We thank the reviewer for their comments, we believe they have helped to substantially strengthen the clarity of the manuscript.

Reviewer: 4 General Comments This is a well-conceived and timely manuscript that applies advanced transcriptomic profiling and machine learning approaches to classify psoriasis into molecular endotypes and predict disease severity. The integration of unsupervised and supervised learning methods to extract biologically meaningful signatures from large-scale datasets demonstrates both technical sophistication and clinical relevance. The manuscript is clearly written, well-structured, and addresses an important unmet need in precision dermatology-namely, the stratification of complex inflammatory conditions like psoriasis. The study is based on comprehensive datasets, and the computational framework is rigorous. Importantly, the authors take appropriate steps to validate their clustering approach and prediction models, including the use of independent cohorts. The methodological transparency and use of SHAP for interpretability are additional strengths. However, a few key areas would benefit from clarification or expansion, particularly with respect to methodological choices and biological interpretation of the identified gene modules.	
Justification of Machine Learning Methods The use of WGCNA and ICA for endotype identification and GPR for PASI score prediction is appropriate and well-executed. However, the manuscript would benefit from a brief justification for choosing GPR over other popular regression models (e.g., Random Forest, XGBoost, or deep learning approaches). Was performance benchmarked across models? This context would help readers assess the robustness and generalizability of the findings.	Thank you for this comment which was also raised by reviewer 3 (point 1). May we refer you to our responses above, our updated methods section (Machine Learning, lines 189-205) and supplementary methods. Additionally, please note that GPR was compared to Ridge models (Supplementary Figures 11-14).
Biological Interpretation of Modules While the identification of co-expression modules and	Thank you for this comment. The supplementary tables 2-5 provide the pathway enrichments and top aligned genes

their association with disease severity is compelling, the biological relevance of these modules could be more deeply explored. Specifically, a brief enrichment analysis or discussion of canonical pathways involved in the major modules would enhance interpretability for translational researchers.	for the WGCNA modules and ICA factors in skin and blood. We have further clarified this in the text: “To further define the functional relevance of the co-expressed WGCNA modules and ICA factors, systems analysis was performed; the top pathway enrichments and exemplar aligned genes are described in supplementary tables 2-5 and in the results below.” (lines 265-267)
Data Harmonization and Batch Effects The manuscript involves multiple public datasets, which may have inherent batch effects. While normalization steps are briefly mentioned, further detail is needed on how batch effects were addressed, particularly across cohorts (e.g., COMBAT, Limma, Harmony). A figure or supplementary table describing cross-dataset comparability would be helpful.	Batch effects across cohorts Please note that no public available data sets were used in this paper. Rather, as described in the manuscript, the PSORT-D and PSORT-R datasets were used for these main analyses:  (i) Derivation of module eigengenes and ICA factor values. WGCNA to identify co-expressed gene modules was carried out only in PSORT-D; module eigengenes were then derived for these modules separately in PSORT-D and PSORT-R, removing the need for batch correction. A similar approach was used for the ICA factor analysis. (ii) Module/factor-trait correlation analysis (figures 2A, 2B and 2E). The correlation of module eigengenes and ICA factor values with traits such as PASI and BMI was carried out separately in PSORT-D and PSORT-R, removing the need for batch correction. (iii) Differential expression (DE) analysis associating PASI with gene expression in skin (figures 3B and 3D; figure 6; figures 7A and 7B). These gene-level DE analyses in skin were carried out using the PSORT-D and PSORT-R data combined. These DE models were implemented using the limma-voom pipeline and so we included cohort as a main effect term in each linear model to control for inter-cohort effects. Please see the disease severity model section of the supplementary methods. We have also added a workbook to the GitHub repository with PCA analysis which shows the effect of cohort on

	transcriptomic variation and how this is mitigated by controlling for cohort in the DE analysis. This is available here: https://github.com/C4TB/PSORT/blob/master/Exploratory_analysis/01_4_DRpca.md (iv) DE analysis associating BMI with gene expression (figure 3). The gene-level DE analysis for BMI was carried out separately in PSORT-D and PSORT-R, removing the need for batch correction. (v) Association between HLA-based patient clusters and baseline PASI. These analyses were carried out using combined PSORT-D and PSORT-R data and so the limma function <code>removeBatchEffect</code> and scaling were used to control for inter-cohort effects. Please see the “Independent component analysis” section of the supplementary methods. In summary, regarding inter-cohort batch effects, most analyses were carried out separately in PSORT-D and PSORT-R, but for integrated analyses we used appropriate methods to control for cohort.
Clinical Translation and Limitations The discussion would benefit from a more nuanced assessment of the clinical applicability and limitations of the proposed molecular subtypes. Are the endotypes stable over time or treatment course? Can these be integrated with other omics or clinical phenotyping tools? Such reflections would broaden the impact of the findings.	Thank you for raising these important questions. To clarify, for both the WGNCA and ICA analysis, samples in lesional and non-lesional across all time points were grouped together into one analysis whereas blood was treated separately. This is further clarified in the results section: “To facilitate comparisons between lesional and non-lesional skin and across time, all skin samples from the adalimumab and ustekinumab drug cohorts at weeks 0, 1 and 12 were analysed together. However, there was extensive transcriptomic variation between skin and blood; therefore, these tissues were analysed separately.” We have also added clarifying text in WGCNA and ICA sections of the main methods, and in the dimensionality reduction section of the supplementary methods.

In the discussion, we have expanded the text of the final paragraph to include:

“Further work is required to derive scalable biomarkers for testing in the clinical setting. One of the advantages of co-expression analysis is dimensionality reduction which facilitates the further development of scalable and clinically applicable biomarkers. Thus, for example, in our on-going work we are investigating the relationship between co-expression modules, clinical traits, a wider range of genetic markers and eQTLs through machine learning. Integrated analysis of samples across drug cohorts, time points, and lesional and non-lesional skin when carrying out WGCNA and ICA greatly reduced the dimensionality of the dataset. However, this approach precluded identification of modules and factors which might be specific to lesional or non-lesional skin, or to certain time points; future work might involve identification of modules and factors separately in these different subsets of samples. Methods such as module preservation analysis for WGCNA, which has been used in this study to compare skin and blood (Supplementary Fig. 4), could then be used to assess the degree to which module co-expression changes across tissues and time points. Crucially, differences in co-expression may be separate to differences in absolute expression levels, which are the focus of our endotype analyses in this paper and more suited to the integrated dimensionality reduction approach taken here. We are planning a separate paper that will examine time point-associated gene expression changes.”

We have also included a short section describing potential limitations of the study: “Potential limitations of the study include lack of randomisation of subjects to treatment arms (although the study cohorts were well-matched) and the absence of treatment adherent assessment (although the response rates seen suggests that participants were adherent to therapy).”

Comment on AI/ML Use The manuscript presents a well-integrated application of AI/ML methods for both unsupervised clustering and outcome prediction. The use of WGCNA and ICA is appropriate for exploring transcriptomic heterogeneity, while Gaussian Process Regression (GPR) effectively models PASI scores with uncertainty quantification. The incorporation of SHAP values enhances interpretability and aligns with current standards for explainable AI in biomedical research. The overall pipeline is thoughtfully designed and adheres to FAIR and reproducible AI principles. While the modelling choices are sound, a brief justification for selecting GPR over other predictive models (e.g., ensemble methods or neural networks) would strengthen the methodological transparency. Overall, the study demonstrates a commendable and responsible use of AI/ML in precision dermatology.	Thank you for this comment, the rationale for selection of GPR was also raised by reviewer 2 and we have responded at length in that section, with a new section in the methods to justify selection of GPR.
Terminology Consistency: Use consistent terminology when referring to endotypes (e.g., cluster labels) and severity strata. Figure Legends: Several figures contain valuable insights but would benefit from more descriptive legends, especially for readers unfamiliar with ICA or GPR output. Supplementary Data: The supplementary materials are comprehensive. However, the authors may consider providing an overview table mapping datasets to analysis steps (e.g., clustering, training, validation). • Ethics Statement: If applicable, include a brief ethics/data access statement clarifying permissions for using patient-level data from public repositories.	Thank you for these comments. We have ensured that consistent terminology for endotypes is applied throughout, for example we change a reference to an adalimumab drug endotype in blood to a disease severity endotype that is specific to the blood of adalimumab treated patients. We have revised all main figure legends to make them more concise and descriptive, particularly to those less familiar with ICA and GPR. Additionally, we have updated Figure 1 to provide a more comprehensive overview of the methodology and a mapping of data sets to the main steps of analysis. An ethics statement is available in the “Prospective observational study” section of the supplementary methods and details about ethical data sharing are available in the “RNA sequencing details” section of the supplementary methods.

	We have modified all legends so that they include: (I) A title with clearer messaging.(II) More methodology details where appropriate.
--	--

Demographic table for all patients in PSORT-D and PSORT-R

		Missing	Overall	Discovery	Replication
			n=146	n=89	n=57
Drug, n (%)	Adalimumab	0	70 (47.9)	41 (46.1)	29 (50.9)
	Ustekinumab		76 (52.1)	48 (53.9)	28 (49.1)
Biologic Naive, n (%)	No	0	35 (24.0)	27 (30.3)	8 (14.0)
	Yes		111 (76.0)	62 (69.7)	49 (86.0)
Ethnicity, n (%)	Asian or Asian British	0	8 (5.5)	6 (6.7)	2 (3.5)
	Black or Black British		3 (2.1)	2 (2.2)	1 (1.8)
	Chinese/Japanese/Korean/Indochinese		1 (0.7)	1 (1.1)	
	White		132 (90.4)	80 (89.9)	52 (91.2)
	Other		2 (1.4)		2 (3.5)
Sex, n (%)	F	0	61 (41.8)	33 (37.1)	28 (49.1)
	M		85 (58.2)	56 (62.9)	29 (50.9)
Psa, n (%)	Negative	0	101 (69.2)	62 (69.7)	39 (68.4)
	Positive		45 (30.8)	27 (30.3)	18 (31.6)
Flexural, n (%)	No	0	77 (52.7)	53 (59.6)	24 (42.1)
	Not Done		4 (2.7)	4 (4.5)	
	Yes		65 (44.5)	32 (36.0)	33 (57.9)
Scalp, n (%)	No	0	24 (16.4)	17 (19.1)	7 (12.3)
	Not Done		4 (2.7)	4 (4.5)	
	Yes		118 (80.8)	68 (76.4)	50 (87.7)
Palms, n (%)	No	0	108 (74.0)	66 (74.2)	42 (73.7)
	Not Done		4 (2.7)	4 (4.5)	
	Yes		34 (23.3)	19 (21.3)	15 (26.3)
Soles, n (%)	No	0	110 (75.3)	67 (75.3)	43 (75.4)
	Not Done		5 (3.4)	4 (4.5)	1 (1.8)
	Yes		31 (21.2)	18 (20.2)	13 (22.8)
Nails, n (%)	No	0	49 (33.6)	25 (28.1)	24 (42.1)
	Not Done		4 (2.7)	4 (4.5)	
	Yes		93 (63.7)	60 (67.4)	33 (57.9)
Age, mean (SD)		0	45.1 (12.4)	44.4 (12.3)	46.2 (12.5)
Age of onset, mean (SD)		2	23.2 (14.1)	22.6 (13.2)	24.1 (15.4)
BMI, mean (SD)		1	31.4 (7.3)	30.8 (6.4)	32.2 (8.5)
wk00 PASI, mean (SD)		0	15.6 (6.6)	15.3 (5.9)	16.2 (7.6)
wk01 PASI, mean (SD)		2	14.0 (6.4)	13.7 (6.0)	14.4 (6.9)
wk04 PASI, mean (SD)		3	9.1 (5.4)	9.1 (5.7)	9.1 (5.1)
wk12 PASI, mean (SD)		0	3.9 (4.7)	3.9 (4.9)	3.9 (4.4)
Cw6, n (%)	Negative	8	73 (52.9)	46 (56.1)	27 (48.2)
	Positive		65 (47.1)	36 (43.9)	29 (51.8)
DeltaPASI, mean (SD)		0	0.7 (0.3)	0.7 (0.2)	0.7 (0.3)
Smoking, n (%)	No	0	48 (32.9)	27 (30.3)	21 (36.8)
	Yes		98 (67.1)	62 (69.7)	36 (63.2)
Alcohol, n (%)	No	0	45 (30.8)	24 (27.0)	21 (36.8)
	Yes		101 (69.2)	65 (73.0)	36 (63.2)

Demographic table for all patients in PSORT-D and PSORT-R, except the 13 patients for whom only skin or blood samples were analysed

		Missing	Overall	Discovery	Replication
			n=133	n=76	n=57
Drug, n (%)	Adalimumab	0	69 (51.9)	40 (52.6)	29 (50.9)
	Ustekinumab		64 (48.1)	36 (47.4)	28 (49.1)
Biologic Naive, n (%)	No	0	34 (25.6)	26 (34.2)	8 (14.0)
	Yes		99 (74.4)	50 (65.8)	49 (86.0)
Ethnicity, n (%)	Asian or Asian British	0	8 (6.0)	6 (7.9)	2 (3.5)
	Black or Black British		2 (1.5)	1 (1.3)	1 (1.8)
	White		121 (91.0)	69 (90.8)	52 (91.2)
	Other		2 (1.5)		2 (3.5)
Sex, n (%)	F	0	56 (42.1)	28 (36.8)	28 (49.1)
	M		77 (57.9)	48 (63.2)	29 (50.9)
PsA, n (%)	Negative	0	91 (68.4)	52 (68.4)	39 (68.4)
	Positive		42 (31.6)	24 (31.6)	18 (31.6)
Flexural, n (%)	No	0	69 (51.9)	45 (59.2)	24 (42.1)
	Not Done		4 (3.0)	4 (5.3)	
	Yes		60 (45.1)	27 (35.5)	33 (57.9)
Scalp, n (%)	No	0	21 (15.8)	14 (18.4)	7 (12.3)
	Not Done		4 (3.0)	4 (5.3)	
	Yes		108 (81.2)	58 (76.3)	50 (87.7)
Palms, n (%)	No	0	97 (72.9)	55 (72.4)	42 (73.7)
	Not Done		4 (3.0)	4 (5.3)	
	Yes		32 (24.1)	17 (22.4)	15 (26.3)
Soles, n (%)	No	0	99 (74.4)	56 (73.7)	43 (75.4)
	Not Done		5 (3.8)	4 (5.3)	1 (1.8)
	Yes		29 (21.8)	16 (21.1)	13 (22.8)
Nails, n (%)	No	0	44 (33.1)	20 (26.3)	24 (42.1)
	Not Done		4 (3.0)	4 (5.3)	
	Yes		85 (63.9)	52 (68.4)	33 (57.9)
Age, mean (SD)		0	44.9 (12.7)	44.0 (12.7)	46.2 (12.5)
Age of onset, mean (SD)		1	23.0 (14.1)	22.2 (13.1)	24.1 (15.4)
BMI, mean (SD)		0	31.4 (7.5)	30.8 (6.6)	32.2 (8.5)
wk00 PASI, mean (SD)		0	15.6 (6.6)	15.1 (5.7)	16.2 (7.6)
wk01 PASI, mean (SD)		1	13.8 (6.4)	13.5 (6.0)	14.4 (6.9)
wk04 PASI, mean (SD)		3	9.0 (5.4)	8.9 (5.7)	9.1 (5.1)
wk12 PASI, mean (SD)		0	4.0 (4.9)	4.0 (5.2)	3.9 (4.4)
Cw6, n (%)	Negative	1	69 (52.3)	42 (55.3)	27 (48.2)
	Positive		63 (47.7)	34 (44.7)	29 (51.8)
DeltaPASI, mean (SD)		0	0.7 (0.3)	0.7 (0.2)	0.7 (0.3)
Smoking, n (%)	No	0	45 (33.8)	24 (31.6)	21 (36.8)
	Yes		88 (66.2)	52 (68.4)	36 (63.2)
Alcohol, n (%)	No	0	42 (31.6)	21 (27.6)	21 (36.8)
	Yes		91 (68.4)	55 (72.4)	36 (63.2)